# MOMSO 1.0 - an eddying Southern Ocean model configuration with fairly equilibrated natural carbon

Heiner Dietze[1,2], Ulrike Löptien[1,2], and Julia Getzlaff[1]

[1]GEOMAR, Helmholtz Centre for Ocean Research Kiel, Düsternbrooker Weg 20, D-24105 Kiel, Germany.
[2]Institute of Geosciences, University of Kiel, Kiel, Germany.

**Correspondence:** H. Dietze
(hdietze@geomar.de)

## Abstract

We present a new near-global coupled biogeochemical ocean-circulation model configuration. The configuration features a horizontal discretization with a grid spacing of less than $11\,km$ in the Southern Ocean and gradually coarsens in meridional direction to more than 200 km at 64° N where the model is bounded by a solid wall. The underlying code framework is GFDL's Modular Ocean Model coupled to the **B**iology **Li**ght **I**ron **N**utrients and **G**asses (BLING) ecosystem model of Galbraith et al. (2010). The configuration is unique in that it features both a relatively equilibrated oceanic carbon inventory and an eddying ocean circulation based on a realistic model geometry/bathymetry - a combination that has been precluded by prohibitive computational cost in the past. Results from a simulation with climatological forcing and a sensitivity experiment with increasing winds suggest that the configuration is sufficiently equilibrated to explore Southern Ocean Carbon uptake dynamics on decadal timescales. The configuration is dubbed **MOMSO** a **M**odular **O**cean **M**odel **S**outhern **O**cean configuration.

## 1 Introduction

The Southern Ocean, also known as the Antarctic Ocean, comprises the southernmost waters of the World Oceans. It's southern boundary is set by the Antarctic Continent. As concerns its northern boundary to the Atlantic, Pacific and Indian Ocean, there is no consensus. One common definition is the location of the Subtropical Front (STF), which separates the relatively saline subtropical waters from the fresher sub-Antarctic waters. This definition is straightforward because it characterizes the transition from one oceanographic regime to another, distinctly different, one. However, because the position of the front varies with time, the definition adds complexity to certain analysis such as oceanic inventories. A pragmatic solution is to define a fixed latitude as the northern boundary. We choose 40°S in order to facilitate a comparison with e.g. Lovenduski et al. (2013) and Dietze et al. (2017).

The atmospheric conditions in the Southern Ocean are characterized by frequent cyclonic storms that travel eastward around the Antarctic Continent and high-pressure areas over the poles. This averages to a strong westerly wind belt between 40°S and 60°S and rather weak and irregular polar easterlies from 60°S on southwards. The predominant low-frequency mode of atmospheric variability is summarized by the Southern Annular Mode (Limpasuvan and Hartmann, 1999; Marshall, 2003) which essentially describes the strength and position of the westerly wind belt.

The strong westerly wind belt drive the Antarctic Circumpolar Current (ACC), the "mightiest current in the oceans" (Pickard and Emery, 1990) which is unobstructed by continents. The ACC circumnavigates the globe zonally and, thus, links the Atlantic, Pacific, and Indian Ocean. A northward Ekman transport (also driven by the westerlies) drives up- and downwelling (Marshall and Speer, 2012) and is part of a meridional overturning circulation. Among the major processes controlling the net
overturning is the so-called "eddy-compensation" which largely cancel changes in wind-driven overturning (c.f. Marshall and Radko, 2003; Viebahn and Eden, 2010; Abernathey et al., 2011, 2016; Hallberg and Gnanadesikan, 2006; Thompson et al., 2014; Tamsitt et al., 2017). Further complexity is added by sea-ice which modulates air-sea buoyancy fluxes and thus exerts control on convection, a prerequisite for Antarctic Bottom Water (AABW) formation.

Being one of the few places world-wide where deep water (such as AABW) is formed, the Southern Ocean plays a major role
in the global carbon budget. Convection events tap into the abyssal ocean and modulate the difference between atmospheric and oceanic $CO_2$ concentrations which drive net air-sea fluxes. A comprehensive quantitative understanding has not been reached yet, but there is consensus that the variability in the extent to which deep-water masses in the Southern Ocean are isolated from the atmosphere, is among the major drivers regulating atmospheric $CO_2$-variability (e.g., Anderson et al., 2009; van Heuven et al., 2014; Ritter et al., 2017). Consequently, the Southern Ocean has shifted into the limelight of climate research (DeVries
et al., 2017; Tamsitt et al., 2017; Langlais et al., 2017, and many more).

To date, the Southern Ocean accounts for almost half of the global oceanic $CO_2$ uptake from the atmosphere (Takahashi et al., 2012). But, there is concern that anticipated climate change (e.g., via changes of atmospheric circulation and sea-ice cover) may trigger substantial changes in the Southern Ocean carbon budget (e.g., Heinze et al., 2015; Abernathey et al., 2016) such that the current rate of uptake may well decline in decades to come. Indications for the existence of such triggers have
been revealed by observation-based atmospheric reanalysis products which show an ongoing strengthening and a poleward shift of the southern westerly winds since the 1970s (Thompson and Solomon, 2002).

This observed trend is projected by climate scenarios to intensify (e.g., Simpkins and Karpechko, 2012) and it is straightforward to assume that the associated wind-driven circulation affects biogeochemical dynamics and, eventually, the oceanic carbon budget. A comprehensive understanding of the link between changing winds and oceanic upwelling of carbon-rich deep
waters (which, in turn, affects surface saturation and net air-sea $CO_2$ exchange) has, however, not been achieved yet. To this end, the role of mesoscale ocean eddies is especially uncertain: the current generation of coarse resolution (non-mesoscale-resolving) models suggests that a poleward shift and an intensification of the Southern Ocean westerlies results in a strengthening of the subpolar meridional overturning cell (e.g., Saenko et al., 2005; Hall and Visbeck, 2002; Getzlaff et al., 2016) and, consequently, in increased upwelling of deep water south of the circumpolar flow, which is rich in dissolved inorganic carbon
(e.g., Zickfeld et al., 2007; Lenton and Matear, 2007; Lovenduski et al., 2008; Verdy et al., 2007). The net effect being here that changes of the atmospheric circulation reduce the capability of the Southern Ocean to sequester carbon away from the atmosphere. Early on, however, there were indications (e.g., Böning et al., 2008; Hallberg and Gnanadesikan, 2006; Hogg et al., 2008; Screen et al., 2009; Thompson and Solomon, 2002) that this model-behavior is a spurious consequence of the underlying eddy parameterization in coarse resolution models which can not afford to resolve mesoscale dynamics explicitly. Supporting

evidence came from Munday et al. (2014) who showed that mesoscale eddies indeed reduce the sensitivity of oceanic carbon sequestration towards changing wind stress in an idealized model.

To date we know that very different state-of-the-art approaches to parameterize eddies yield surprisingly similar sensitivities of oceanic carbon-inventories to changing winds Dietze et al. (2017). Further it has bee demonstrated that "the conversion of dense waters back to light waters is sensitive to the degree of parameterization of mesoscale eddies in models" (Spence et al., 2010, 2009). The question, however, as to how a realistic (as opposed to featuring an idealized geometry) high resolution coupled biogeochemical ocean circulation model which actually resolves eddies and carbon dynamics explicitly compares to a coarse resolution model hat not been answered yet. Amon the reasons is the prohibitive computational cost that is associated to equilibrating simulated dissolved inorganic carbon concentrations at depth.

In this model-description paper we present the realistic high-resolution model configuration MOMSO 1.0. The configuration features a realistic topography and simulated levels of eddy kinetic energy which do not undercut observed values. This suggests that the configuration explicitly resolves a substantial part of mesoscale-related variability rather than relying on parameterizing their effect. MOMSO is designed to explore the sensitivity of the Southern Ocean carbon uptake to atmospheric changes on decadal scales. The configuration is rendered feasible by recent advances in compute hardware and from its similarity with a spun-up coarse resolution model which delivered the initial conditions for the biogeochemical module. (Note that module refers to the algorithmic entity that calculates the local biogeochemical sources and sinks of the prognostic biogeochemical variables such as nutrients and carbon. The algorithmic entity comprises both the underlying partial differential equations and their approximated representation within the numerical solver.) More specifically, we will showcase that the "level-of-equilibration" of simulated dissolved inorganic carbon allows us to test the sensitivity of the Southern Ocean carbon budget to anticipated climate change patterns.

In summary, this model-description paper aims to (1) describe a new eddying coupled ocean-circulation biogeochemical model configuration of the Southern Ocean, and (2) to outline research questions for which MOMSO may serve as the base of a tool bench.

The project is dubbed **MOMSO**, a configuration of GFDL's **M**odular **O**cean **M**odel version 4p1 with enhanced resolution in the **S**outhern **O**cean. The naming is an homage to the underlying framework, the MOM4p1 release of NOAAs Geophysical Fluid Dynamics Laboratory (GFDL) Modular Ocean Model (Griffies, 2009). The ocean circulation model is coupled to a sea-ice model and the biogeochemical module BLING from Galbraith et al. (2010).

## 2 Model Setup

This model-description paper describes simulations with the Modular Ocean Model (MOM), version MOM4p1 (Griffies, 2009). The configuration is near-global, bounded by Antarctica and $64°$N. In the Southern Ocean the horizontal resolution is higher than $11\,km$ up till $40°$S. The meridional grid resolution coarsens towards the North (Fig. 1). There are no open boundaries and there is no tidal forcing.

The biogeochmical model BLING, short for **B**iology **Li**ght **I**ron **N**utrients and **G**asses (Galbraith et al., 2010) is coupled online

to the ocean-sea ice model. Atmospheric $CO_2$ concentrations are prescribed to a preindustrial level of $278\,ppmv$. The respective carbon inventories and fluxes are referred to as *natural carbon*.

## 2.1 Grid and Bathymetry

The underlying bathymetry is ETOPO5 (c.f. Data Announcement 88-MGG-02, Digital relief of the Surface of the Earth. NOAA, National Geophysical Data Center, Boulder, Colorado, 1988). Using a bilinear scheme the bathymetry is interpolated onto an Arakawa B-grid (Arakawa and Lamp, 1977) with $2400 \times 482$ tracer grid boxes in the horizontal. The ocean-circulation and the sea-ice model share the same horizontal grid.

The vertical discretisation comprises a total of 55 levels. Fig. 1 shows the nominal depth and thickness of each level. The model

bathymetry is smoothed with a filter similar to the Shapiro filter (Shapiro, 1970). The filter weights are 0.25, 0.5 and 0.25. The filtering procedure can only decrease the bottom depth, i.e. essentially, it fills rough holes. The filter is applied three times consecutively because we found this to be a good compromise between unnecessary smoothing on the one hand and numerical instabilities introduced by overly steep topography on the other hand in other high-resolution model configurations (Dietze and Kriest, 2012; Dietze et al., 2014). The resulting bathymetry contained lakes which we filled after visual inspection. In addition,

we filled narrow inlets which had a width of less than three grid boxes. In total, MOMSO has 42 429 759 wet tracer grid boxes. We use the *zstar* coordinate (Stacey et al., 1995; Adcroft and Campin, 2004) in the vertical which is essentially an extension of the nonlinear free surface method of (Campin et al., 2004) to all model levels. The algorithm is renown for its very accurate conservation of tracers. *zstar* ($z^\star$) is calculated as a function of nominal depth ($z$), water depth ($H$) and the free sea surface height ($\eta$) which varies with time:

$$z^\star = H\left(\frac{z - \eta}{H + \eta}\right) \tag{1}$$

(equation 6.6 in Griffies, 2009). The approach overcomes the problem with vanishing surface grid boxes which appears in generic z-level discretisation when sea surface height variations are of similar magnitude than the thickness of the uppermost grid box.

## 2.2 Ocean Circulation

MOM4p1 is a z-coordinate, free surface ocean general circulation model which discretizes the ocean's hydrostatic primitive equations on a fixed Eulerian grid. The vertical mixing of momentum and scalars is parameterized with the K-Profile-Parameterization approach of Large et al. (1994) with the same parameters applied in eddy-permitting global configurations of Dietze and Kriest (2012); Dietze and Löeptien (2013); Dietze et al. (2014), and Liu et al. (2010). The relevant parameters are (1) a critical bulk Richardson number of 0.3 and (2) a constant vertical background diffusivity and viscosity of $10^{-5}\ m^2/s$.

The background values apply also below the surface mixed layer throughout the water column. Both parameterizations of the nonlocal and the double diffusive (vertical) scalar tracer fluxes are applied.

We apply a state-dependent horizontal Smagorinsky viscosity scheme (Griffies and Hallberg, 2000; Smagorinski, 1963, 1993)

to keep friction at the minimal level necessiated by numerical stability. We apply a respective coefficient that sets the scale of the Smagorinsky isotropic viscosity to 0.01. Note that this value is much smaller than the range between 2 to 4 recommended by Griffies and Hallberg (2000) for stability reasons. The ratio behind our choice is to minimize friction which has little physical justification while, at the same time, has been shown to degrade the performance of ocean circulation models (e.g. Jochum et al., 2008).

We use the PPM advection scheme (Colella and Woodward, 1984) for active tracers because it has been shown to perform well in treating sharp gradient in small-scale circulation (e.g. Carpenter et al., 1989). For the biogeochemical tracers we use a flux-limited scheme following Sweby (1984) because we experienced problems in other configurations with negative values for biogeochemical with non-flux-limited schemes in the past. The choice of two different advection schemes may have repercussions. E.g. Lévy et al. (2001) has shown that the choice spuriously affects biogeochemistry in regions of sharp vertical macronutrients gradients such as in oligotrophic regions. We expect that this is not so much an issue in MOMSO because the Southern Ocean vertical nutrient gradients are smoother. Further, we argue that all advection schemes introduce spurious errors (either of diffusive or dispersive nature) that are directly related to the respective property gradient in flow direction. Since all properties (such as temperature, salinity and biogeochemical variables such as nutrients, oxygen and dissolved organic carbon) feature differing gradients they are affected to a differing degree by the spurious behavior of the advection scheme. Hence, even when using the same advection scheme for all properties, the actual transport of properties is inconsistent in that they are all affected by differing degrees by spurious errors. This inconsistency can be greater than the inconsistency introduced by switching from one advection scheme to another.

We do not apply an explicit horizontal background diffusivity other than the contribution that is implicit to the advection scheme.

Several decades into the spin-up the configuration became unstable in coarsely-resolved places where strong currents met rough topography. This may well have been the result of our very low choice of the Smagorinsky isotropic viscosity (as reviewer 1 suspected). Our fix to the instability problem was to set an additional horizontal isotropic Laplacian Viscosity of $600\,m^2/s$ from 10°S to 50°N, of $1200\,m^2/s$ above 50°N and $1800\,m^2/s$ and above 60°N until the northern boundary of the model domain kept the respective oscillations in check. In addition we added Laplacian Viscosity at the exit of Drake Passage (Fig. 2).

## 2.3 Sea Ice

The ocean component is coupled to a dynamical sea ice module, the GFDL Sea Ice Simulator (SIS). SIS uses elastic-viscous-plastic rheology adapted from Hunke and Dukowicz (1997). In the standard version, the simulated sea ice impacts sea surface height. This led to a vicious cycle at some places where sea ice attracts ever more sea ice resulting in unrealistic anomalies in sea surface height and finally in a numerical blow-up of the simulation. Because of the associated computational cost we did not investigate thoroughly into this but - instead - switched to levitating sea ice by deleting line 1353 in ice_model.F90 which reads: $Ice\%p\_surf(i,j) = Ice\%p\_surf(i,j) + grav * x(i,j)$. Note that levitating sea ice is also used e.g. in Hordoir et al. (2019) which is based on the NEMO framework (Hordoir personal communication, 2019).

## 2.4 Biogeochemistry

In our setup, the ocean component is coupled to the BLING ecosystem model of Galbraith et al. (2010). BLING is a prognostic model that, in the basic version, explicitly resolves only four biogeochemical tracers: dissolved inorganic phosphorous, dissolved organic phosphorous, dissolved iron and dissolved oxygen. In this model-description paper we use BLING in conjunction with a carbon module that explicitly resolves dissolved inorganic carbon and alkalinity as described, e.g., in Bernadello et al. (2014).

The design idea behind the "reduced-tracer" model BLING is, on the one hand, to reduce the number of prognostic tracers that are actually advected by the ocean circulation model to a minimum while, on the other hand, to explicitly resolve the most influential environmental conditions that control the net biotic carbon uptake (macro- and micronutrients phosphorous and iron; oxygen which controls heterotrophic respirartion ). The ratio behind reducing the number of prognostic tracers is that even simple models (resolving macronutrients, two phytoplankton species, zooplankton and detritus) are hard to constrain with available observations already (c.f. Löptien and Dietze, 2015, 2017, 2019). As a side effect, the reduced number of prognostic tracers reduces the computational cost.

## 2.5 Initial Conditions and Spin-up Procedure

The circulation model starts from rest (i.e. initial velocities are nil) with initial values for temperature and salinity taken from WOA2009 (Locarnini et al., 2010; Antonov et al., 2009, respectively). After 20 years of physics-only spin-up, the biogeochemical model is hooked on. In contrast to the physics we do not use an observational product to initialize the prognostic variables of the biogeochemical model also because the observations of dissolved iron are so sparse that Orr et al. (2017) suggest not to use them for model initialization. Instead we interpolate the initial conditions for the biogeochemical tracers are interpolated from the fully spun-up coarse resolution configuration used by Dietze et al. (2017) (their "FMCD" simulation) which is, apart from the spatial discretization (and related parameters in the physical parameterizations of unresolved processes), identical to the MOMSO configuration. A beneficial aspect of this procedure is that it accelerates the equilibration of the carbon dynamics substantially compared to using observational products. After a subsequent 60 year-long spin-up with on-line biogeochemistry the model allows already (as we will put forward in Section 4) for an investigation of circulation-driven decadal changes of the Southern Ocean Carbon Budget.

## 2.6 Boundary Conditions and Sponges

The boundaries towards the Arctic (i.e. the northern end of the model domain shown in Fig. 1) are closed (i.e. they are represented by solid and flat walls). Temperature and salinity are restored to climatological annual mean estimates (Locarnini et al., 2010; Antonov et al., 2009) in so-called sponge zones located in the coarse-resolution domain. The sponge zones along with restoring timescales are shown in Fig. 3. The purpose of these sponges is to ensure realistic deep-water characteristics even though northern-hemisphere deep-water formation processes are handicapped by the combination of coarse resolution with the absence of eddy-parameterizations.

At the air-sea boundary we apply climatological atmospheric conditions taken from the Corrected Normal Year Forcing (COREv2 Large and Yeager, 2004). In addition we apply a surface salinity restoring to climatological values (Antonov et al., 2009) with a timescale of 1/2 year throughout the model domain.

Atmospheric $CO_2$ concentrations are prescribed to a preindustrial level of $278\,ppmv$. Thus the simulated oceanic carbon is also referred to as *natural carbon*. Biogeochemical air-sea fluxes (of iron) are identical to the ones applied in Galbraith et al. (2010) and Dietze et al. (2017).

This paper refers to output from two simulations dubbed REF and WIND. Both simulations share the same 80 year spin-up described in Sect. 2.5. WIND branches off from REF during the nominal year 1980 and is exposed to ever increasing wind speeds south of $40°S$. The increase is linear at a rate of 14% in 50 years, consistent with results from a reanalysis of the period 1958 to 2007 (Lovenduski et al., 2013). The ratio behind presenting results from the sensitivity experiment WIND in this model description paper is that they serve as a reference point against wich the remaining model drift in REF (which is there since we can not afford thousand of years of spinup) can be compared with. Based on this comparison Sec. 4 sketches - very briefly - research questions that may be tackled with MOMSO.

IO-related hardware problems caused data loss. REF covers the period from 1980 to 2024, WIND covers 1980 to 2022 only. Please note that a comprehensive analysis of the sensitivity experiment WIND is beyond the scope of this model-description paper. Here we simply want to show that REF is sufficiently equilibrated so that trends effected by decadal-scale changes in winds can clearly be distinguished against the background trend that is still persistent.

## 3  Results

In the following we evaluate our model (simulation REF) by comparing our climatological results from the nominal years 1980 - 2024 to observational data. One problem is the tradeoff between data density and the length of the period the data is representative for. For any given year data densities are insufficient to compile a comprehensive 3-dimensional gridded data product. Binning data of several years into one product closes spatial data gaps, but then, this blurs the referencing to an ever (anthropogenically-driven) changing system state. This problem is especially pronounced in the Southern Ocean where in-situ data acquisition is complicated by hostile environmental conditions.

The climatological atmospheric boundary conditions which drive our ocean model are representative for the period 1958-2000. Climatological data products are typically biased in that they contain more recent data being the result of recent technological advances (such as the development of autonomous platforms). Hence, a model evaluation is not straightforward and it is difficult to define meaningful model-data misfit metrics.

A potential application for the model configuration presented here is to explore the role of mesoscale features, or, eddies in determining the $CO_2$-uptake of the Southern Ocean. One hypothesis this model is set-up to test is whether spatially-unresolved dynamics in IPCC-type coarse resolution models biases their carbon uptake sensitivity. In order to come to a meaningful conclusion on this, our high-resolution model has to perform with a fidelity similar or superior to that of IPCC-type coarse resolution models. In the following we list and explain our choice of model assessments which we deem relevant in this

respect. Wherever applicable we will follow the approach taken by Sallée et al. (2013) and Sallée et al. (2013) who assessed the performance of CMIP5 models and Russell et al. (2018) who suggested metrics for the evaluation of the Southern Ocean in coupled climate models and Earth System Models.

Please note, that we refrain from using state estimates based on data assimilation such as B-SOSE Verdy and Mazloff (2017) for biogeochemical parameters. On the one hand these state estimates are especially useful in the Southern Ocean where data is sparse. On the other hand these estimates rely - especially when data is sparse - on the realism of the underlying ocean-circulation biogeochemical model framework. In the case of B-SOSE the underlying model framework is so similar to our model configuration (high resolution ocean circulation model with Galbraith et al. (2010) biogeochemical model) that we essentially would assess the benefit of the assimilation scheme in fitting observational data rather rather than assessing our model configuration.

The model-data comparison is divided into the following sections:

- **Ocean circulation** (Sec. 3.0.1) which, e.g., effects the transport of carbon-rich deepwater to the surface, shapes the locations of fronts and constitutes a major pathway for nutrients that are essential for phytoplankton growth. For the evaluation of surface currents, we use exemplary snapshots showcasing main circulation paths and spatial variability along with climatological sea surface height which is indicative for the barotropic circulation. More quantitative measures of transport characteristics are provided for the ACC (Drake Passage transport) and the Southern Ocean meridional overturning circulation. Further, we assess the strength of the cyclonic Ross and Weddell polar gyres which are formed by interactions between the ACC and the Antarctic continental shelf. Special emphasis is here on the larger Weddell Gyre. This gyre entrains heat and salt from the ACC and carries them to the Antarctic continental shelves, where deep and bottom waters are produced and, thus, establish an intermittent connection between the atmosphere (surface ocean) and the deep oceanic carbon (nutrient) pool.

- **Eddy kinetic energy** (EKE, Sec. 3.0.2), which is an important measure for the mesoscale activity and thus a key proxy for realistically reproducing eddy-dynamics. At the surface the EKE can be derived from the variability of the sea surface height (SSH), a measure that can be directly observed from space by satellite altimetry.

- **Surface mixed layer depth** (MLD, Sec. 3.0.3) The MLD is determined by (1) the stability of the water column (i.e. the vertical density gradient), (2) wind-induced turbulence which provides energy for eroding the stability and thereby deepens the MLD, and (3) air-sea buoyancy fluxes which can, depending on their sign, increase or decrease the stability of the water column and thus shallow or deepen the MLD. The MLD is an important concept or metric because it locates that fraction of the upper water column that is in direct contact with the atmosphere and sets air-sea gradients of temperature and partial pressure of gases such as carbon dioxide or oxygen. Further, changes in MLD modulate the level of average light levels experienced by phytoplankton: deep MLDs define conditions where phytoplankton spend more time mixed downwards away from the sun-lit surface which reduces their growth and biotic carbon uptake. Antagonistic to this light deprivation, however, deepening MLDs are typically associated to transports of nutrients essential for phytoplankton from depth to the sun-lit nutrient-depleted surface.

- **Temperatures** (Sec. 3.0.4) influences the rate of biological turn-over, affects the solubility pump of carbon and the density of sea water. Cooling typically results in a reduction of buoyancy and causes convection. Spatial temperature gradients are associated to geostrophic circulation (if they are not salinity-compensated). Vertical temperature gradients affect the stability of the water and precondition vertical mixing. The sea surface temperature (SST) can be directly observed from space and, therefore, is available in an unrivaled (compared to in-situ measured properties) spatial and temporal resolution. Temperatures at depth are important for basal melting and, thus, are related to the formation rate of Antarctic Bottom Water (AABW). AABW formation, in turn, affects the solubility and biotic pump of carbon.

- **Salinity** (Sec. 3.0.5) is related to the density of sea water. Saltening by brine rejection can cause convection, meltwater on the other hand can form buoyant lenses thus increasing the local stability of the water column which prevents vertical mixing. Spatial salinity gradients are associated to geostrophic circulation (if they are not temperature-compensated). Vertical salinity gradients affect the stability of the water and precondition vertical mixing.

- **Sea Ice** (Sec. 3.0.6) caps the direct exchange between atmosphere and ocean and thus controls the air-sea gas exchange of $CO_2$. It also modulates the air-sea buoyancy forcing by, e.g., insulating the surface from heat loss or by brine rejection during ice formation. Further, it shields the surface water from solar irradiance and hampers the assimilation of $CO_2$ by autotrophic plankton.

- **Nutrients** (Sec. 3.0.7) which are essential to the growth of autotrophic plankton and whose availability exert major control on the biological pump. The most important macronutrient is bioavailable phosphorous such as phosphate ($PO_4$) because its availability is essential to all phytoplankton (and cyanobacteria). The distribution of $PO_4$ is determined by the interplay of ocean circulation transporting $PO_4$ dissolved in sea water and marine biota which utilize phosphorous to build biomass (typically at the surface) and release $PO_4$ in the course of degradation of organic material (typically at depth). In addition we assess simulated iron concentrations since, the Southern Ocean is well-known for being a site where this is limiting the growth of autotrophs (e.g. Boyd and Ellwood, 2010). Further, in order to guide the interpretation of simulated $PO_4$ concentrations at depth we include a meridional section of dissolved oxygen in this section: oxygen is reset to values close to saturation at the surface while its sinks in the interior are typically assumed to be linearly linked to interior sources of $PO_4$ (which is not reset at the surface). Hence, cases where biases in simulated $PO_4$ concentrations are inconsistent with biases in simulated oxygen concentrations contain information on what exactly within the interplay of ocean circulation and biogeochemistry is the cause for the simulated biases.

### 3.0.1 Ocean Circulation

Fig. 4 shows simulated climatological sea surface height (SSH) along with an estimate based on observations from space (MADT, AVISO). The model captures the main features of the global barotropic circulation, even in the coarsly-resolved domain north of $40°$S, such as in the subtropical gyres in both hemispheres north and south of the equator (positive SSH anomalies) and in the subpolar gyres around $50°$N and $60°$N in the Pacific and Atlantic (negative SSH anomalies), respectively.

Further, as indicated by he transition from yellow to blue in the Southern Ocean in Fig. 4 the model reproduces the northern boundary of the Antarctic Circumpolar Current (ACC), also referred to as the northern edge of the Subantarctic Front (SAF).

A quantitative numerical comparison of observed (AVISO) and simulated climatologic surface speeds yields a spatial correlation of 0.6 in the high-resolution domain south of $40°$N. The spatial variance in the speed of simulated surface currents is typically 30% higher than the observational estimate and the respective simulated mean is 20% higher than in the observations. Apparently, a (yet-to-be-quantified) fraction of this misfit is attributable to errors in the estimates from space (c.f. Fratantoni, 2001).

Fig.5 shows exemplary snapshots of the surface velocities as observed from space (left, MADT AVISO) and simulated (right, MOMSO). The boundary conditions of the model are not identical to the atmospheric boundary conditions on that specific day and the highly non-linear characteristics of eddy-dynamics renders an "eddy-to-eddy" similarity without data assimilation impossible. So the purpose of Fig.5 is to demonstrate the similarity of patterns and the major transport pathways in the high-resolution domain south of $40°$N. The closeup into the Agulhas retroflection zone (Fig. 6) which is located right in the transition zone from very high to very coarse resolution (c.f. Fig. 1 c) highlights that the realism of simulated patterns is sustained right up into the transition to coarser, non-eddy-resolving resolution.

Fig. 7 features the simulated transport through the Drake Passage. (Here we refer to the reference simulation only; simulation WIND is discussed in Section 4.) The simulated Drake Passage transport averages at 99 Sv (nominal years 1980–2023). This is biased low compared to observational estimates that range from 110 to $170\,Sv$ (Withworth (1983) for 1979; Cunningham et al. (2003) for 1993–2000; Chidichima et al. (2014) for 2007–2011). Similar biases have been reported in other high-resolution configurations and may, according to Dufour et al. (2015), be related to a deficient representation of the overflow of dense waters, formed along the Antarctic coasts. If so, the ACC bias may well be endemic to z-level models which struggle to represent complex topography (in comparison to more elaborate numerical approaches such as, e.g., finite elements). A comprehensive investigation is beyond the scope of this manuscript. But, still, the problem is an intriguing one – especially since, historically, (coarse resolution) models started out from an opposing bias dubbed Hidaka's Dilemma (Hidaka and Tsuchiya, 1953), where an excessive ACC transport could, only by application of unrealistically high friction, be fenced into realistic bounds.

In terms of Eulerian meridional overturning in the Southern Ocean our model values are consistent with the Southern Ocean State Estimate of Mazloff et al. (2010): Mazloff et al. (2010) find a surface meridional overturning cell across $32°$S of $12 \pm 12\,Sv$ and an abyssal cell of $13 \pm 6\,Sv$. We find a climatological mean value for the upper cell of $12 \pm 4\,Sv$ and $8 \pm 3\,Sv$ for the lower cell. Please note that more elaborate measure such as the physically more meaningful overturning calculated on density coordinates or its approximation given by the residual mean streamfunction (c.f. McIntosh and McDougall, 1996; Viebahn and Eden, 2010) are beyond the scope of this manuscript which is intended to serve as a model-description only.

In terms of transports of the Weddell and Ross gyre our simulation is slightly biased high. In the Weddell gyre we simulate $70\,Sv$ while published estimates range from $40 \pm 8\,Sv$ (Southern Ocean State Estimate, Mazloff et al., 2010) to $56 \pm 10\,Sv$ (recent SODA estimate Yongliang et al., 2016) 55.9 $\pm 9.8\,Sv$ and $61 - 66\,Sv$ (Schröder and Fahrbach, 1999). In the Ross gyre we simulate $35\,Sv$ while published estimates range from $20 \pm 5\,Sv$ (Mazloff et al., 2010) to $15 - 30\,Sv$ (Chu and Fan, 2007) and $37 \pm 6\,Sv$ (Yongliang et al., 2016).

### 3.0.2 Eddy kinetic energy and sea surface height

Fig. 8 shows that the simulated climatological EKE reproduces the amplitudes and spatial patterns observed from space (MSLA AVISO data) as can be expected from high-resolution configuration (e.g. Delworth et al., 2012; Barnier et al., 2006). This applies especially in the high-resolution region south of $40°$S where typical deviations in simulated EKE are rather small ($< 0.03\,m^2/s^2$) compared to the maximum values found in the ACC ($> 0.3\,m^2/s^2$, c.f. Fig. 9). The correlation coefficient between simulated and observed EKE patterns is 0.7. The simulated EKE features 7% more spatial variance and an average of 8% more energy within the high-resolution domain south of $40°$S. By averaging all simulates values with their immediate spatial neighbors 2 times consecutively the simulated variance levels become similar to observed values and the respective correlation increases to 0.71. We interpret this as an indication that the model resolution is higher than the effective spatial resolution of the MSLA AVISO data. This may also explain why simulated EKE amplitudes are generally higher than estimates from space (c.f. Fratantoni, 2001).

We conclude that we found no evidence of an underestimation of simulated mesoscale activity. This, in turn, implies that our combination of spatial resolution and (parameterization of) friction in the Southern Ocean is suitable to allow for investigations into eddy-driven processes.

### 3.0.3 Surface mixed layer depth

MLD is a concept that locates that fraction of the upper water column that is well mixed. MLD is typically derived from vertical profiles of temperature and salinity rather than from direct measurements of mixing intensity (i.e. turbulence). As a consequence algorithms calculating MLDs struggle with distinguishing between identifying actively mixed layers and those which have been actively mixed in the past but are no longer fed by energy fueling actual turbulence and associated mixing. The first three columns of Fig. 21 provide a comparison between contemporary (in the sense that they are still used in current peer-reviewed literature) databases and algorithms to compute MLD: CARS 2009 distributed by CSIRO http://www.marine. csiro.au/~dunn/cars2009/ and introduced by Condie and Dunn (2006) is predominantly based on historical CTD observations while both Argo Al. and Argo Thresh. are based on data from Argo floats. The difference between Argo Al. and Argo Thresh. is the algorithm used to derive MLDs Holte et al. (2017). We find a high correlation between the different algorithms Argo Al. and Argo Thresh. in the region south of $40°$S (0.98 and 0.91 in summer and winter, respectively). The correlation between one database to another (i.e. Argo vs. CARS2009) is much less ($\approx 0.7$ for both winter and summer - irrespective of algorithm). Argo Thresh. is closest to our simulated patterns (4th row in Fig. 21) and we find a respective correlation of 0.5 and 0.6 in summer and winter, respectively.

In summary (c.f. winter statistics in Fig. 22) we find that the variance in simulated MLDs is higher than in the observations and that the differences in correlation between one MLD product to another are comparable to our model-observation misfit. In terms of bias we find that our simulated winter MLD is, on average, $18\,m$ deeper than Argo Thresh. This is a major improvement compared to (coarse resolution and fully-coupled) CMIP5 models which feature similar correlations but winter mixed layer depths biased low by typically by $50 - 100\,m$ (Sallée et al., 2013, their Fig. 4a).

### 3.0.4 Temperature

Fig. 10 shows a comparison of the simulated SST with WOA09 (Locarnini et al., 2010) and ARGO float-based (Roemmich and Gilson, 2009) observations. The overall pattern is well reproduced - an exception being a local unusual high bias of about $3°C$ close to the Antarctic coast (between $120°E$ and $160°E$). Averaged over the Southern Ocean (Fig. 11) the simulated SSTs are within the observational range (although at at the lower edge) of the years 1960 to 2010 (Fig. 12 calculated from HadISST, Rayner et al., 2003).

Fig. 17 summarizes the fidelity of simulated temperatures in the interior. Following Sallée et al. (2013) we discuss the five major water masses (subtropical water, mode water, intermediate water, circumpolar deep water, bottom water) separately:

Subtropical water (TW): MOMSO features a slight cold bias at the surface (less than $0.5°C$) and a warm bias of $\approx 1°C$ in the interior (Fig. 17). This is apparently very good when compared against the $2.7°$ warm bias in the CMIP5 models (Sallée et al., 2013). There are, however indications that the $2.7°$ bias is a consequence of biased air-sea heat fluxes in the Southern Hemisphere in the coupled models rather than being the consequence of an unrealistic oceanic circulation.

Mode water (MW): The subduction of surface waters is dominated by mode water in the Southern Ocean. Hence, MW is an important agent in sequestering anthropogenic carbon away from the atmosphere. MOMSO's MW is too warm by $\approx 1°C$ which compares favorably to the much larger bias of an ensemble mean bias of $3.5°C$ in CMIP5 models (Sallée et al., 2013).

Intermediate Water (IW): IW, is denser type of MW which ventilates the thermocline and is sequestered away from the atmosphere longer than MW because it protrudes deeper into the water column. MOMSO's IW is biased warm by $\approx 1.1°C$ (Fig. 17) which compares favorably against the $3.5°C$ warm bias in CMIP5 models (Sallée et al., 2013).

Circumpolar Deep Water (CDW): During its formation at the surface CDW taps into the abyssal waters rich in natural carbon. Hence it plays a key role for the oceanic sequestration of natural carbon dioxide (e.g. Le Quéré et al., 2009). MOMSO overestimates the CDW temperature by $\approx 1°C$. This is substantially more than the $1°C$ overestimation of the CMIP5 ensemble mean but still within the envelope of the ensemble wich peaks at an overestimation of $1.5°C$ (Sallée et al., 2013).

Antarctic Bottom Water (BW): MOMSO overestimates BW temperatures by $1°C$ while the CMIP5 ensemble mean compares more favorably with an underestimation of only $0.4°C$. These biases are consistent with MOMSO featuring too much sea ice coverage which caps the ocean from the cooling atmosphere c.f. Sec. 3.0.6). Likewise CMIP5 models tend to underestimate sea ice coverage (Turner et al., 2013) which may overly expose the ocean to the cooling atmosphere and thereby cause the respective cold bias in BW temperature.

### 3.0.5 Salinity

Fig. 14 shows a comparison of the simulated climatological mean sea surface salinity (SSS) with three observation-based products (WOA09 (Antonov et al., 2009), ARGO (Roemmich and Gilson, 2009) and SMOS (Köhler et al., 2015)). Within the spatially highly-resolved Southern Ocean the simulated sea surface salinity is in good agreement with the observations. Towards the north, where the resolution coarsens the model fidelity disintegrates; particularly in the Atlantic, and Indian Sector the model is biased low.

Averaged over the Southern Ocean the simualted SSS is biased low (compare reference in Fig. 15) compared to recent observational estimates during 2005 – 2017 (Fig. 16). Fig. 15 suggests that increasing winds can increase simulated SSSs up to observed values. This could be interpreted as a mismatch between climatological forcing and observation period. (Please note that a comprehensive analysis is beyind the scope of this model-description paper.) Fig. 15 suggests further, that the surface salinity restoring (c.f. Section 2.6) is weak enough to allow for substantial SSS dynamics in response to e.g. increasing winds.

Fig. 18 summarizes the fidelity of simulated temperatures in the interior. Following Sallée et al. (2013) we discuss the five major water masses (subtropical water, mode water, intermediate water, circumpolar deep water, bottom water) separately:

Subtropical water (TW): MOMSO is biased low by 0.5 at the surface and this makes the simulated TW too light in comparison with WOA09 observations. Sallée et al. (2013) find a smaller ensemble mean bias of 0.2 in the CMIP5 models which is - in contrast to MOMSO - enhanced, in terms of biasing the density towards smaller values, by a 2.7°C warm bias.

Mode water (MW): MOMSO is biased low by 0.1 at the surface and high by 0.1 in the interior. This is in line with the overall poor performance of CMIP5 models in representing the MW (Sallée et al., 2013). Coincidentally, the ensemble mean over all the CMIP5 models averages to a very small fresh bias of 0.02.

Intermediate Water (IW): MOMSO is, again, biased low by 0.1 at the surface and high by 0.1 in the interior. This dampens the equatorward protruding of IW's salinity-minimum signature. Compared against the high bias of only 0.008 in the ensemble mean over all the CMIP5 models MOMSO's bias appears substantial. Compared, however, agains individual CMIP5 models whose characteristics vary widely (Sallée et al., 2013) MOMSO is comparable (i.e. within the envelope spanne by the CMIP5 models).

Circumpolar Deep Water (CDW): MOMSO is biased high by 0.07 south of 75°S at the surface and less than 0.02 equatorwards. These small biases are well within the range set by the CMIP5 models and of the same order of magnitude than the bias of 0.02 of the CMIP5 ensemble.

Antarctic Bottom Water (BW): MOMSO's bias is identical to the high bias of 0.06 in the CMIP5 ensemble mean (Sallée et al., 2013) and much better than the up to 0.5 biases of many individual CMIP5 members. The bias is consistent with the bias in sea ice: an overestimated ice production releases too much brine and shields the ocean from atmospheric cooling. This drives spuriously elevated salinities and temperatures.

### 3.0.6 Sea Ice

Figure 23 shows a comparison of the simulated number of sea ice-covered months per year with an observation estimate (HadISST Rayner et al., 2003). Overall the agreement is good with the following exception: (1) The Weddell and Ross Sea the ice coverage is underestimated by two months. We speculate that this triggers elevated air-sea momentum fluxes and, eventually, biases the respective gyre strengths high (c.f. Section 3.0.1). (2) Overall the simulated ice extent is biased high (compare Fig. 24, black line to Fig. 25). Following Russell et al. (2018) we compare, in addition, against the fractional sea-ice coverage obtained from the National Snow and Ice Data Center (NSIDC, ftp://sidads.colorado.edu/pub/DATASETS/NOAA/G02202_v2/)). The comparison in Fig. 26 confirms the impression that the model overestimates sea ice coverage. Fig. 24 suggests that increasing the wind speeds to levels being more representative for the time period of observations shown in Fig. 25 alleviates this model

bias. In addition Fig. 26 reveals that the model bias features a seasonality with underestimation in January till March and an overestimation of ice coverage from Mai till December. Ranked against the current generation of IPCC models MOMSO fits well into the envelope of the ensemble reported by Turner et al. (2013).

### 3.0.7 Nutrients

Simulated Southern Ocean $PO_4$ surface concentrations are biased low by down to $0.6 \ mmol \ P/m^2$ locally (Fig. 27 a, c). The reason is not straightforward to identify because it could be associated to a deficient physical module, a deficient biogeochemical module, or both. In the following we will present an indication that the problem is associated to a deficient formulation of iron limitation, argue why the formulation of light-limitation is unlikely to be the main problem and put the model-data misfit into perspective.

The uptake of $PO_4$ at the sun-lit surface by autotrophic phytoplankton is known to be limited by the availability of light and the availability of the micronutrient iron (in the Southern Ocean). Fig. 29 features a comparison of simulated iron concentrations with observations. Even though the spatial and temporal coverage of iron measurements is still sparse the emerging pattern is one where the simulated biotic iron-drawdown at the surface appears to be too strong. Surface iron concentrations are biased low, just like the $PO_4$ concentrations and they appear so throughout an annual cycle. Such deficient model behavior can be

caused by insufficient throttling of phytoplankton growth by both iron and light limitation. Looking closer in seasonal model-data misfits, however, suggests that a deficient iron limitation is more likely to be the cause: The dependency of growth (and associated micro- and macronutrient drawdown at the surface) is known to be a highly nonlinear function of environmental drivers. We find that the bias in surface $PO_4$ concentrations is almost constant over the course of a seasonal cycle (Fig. 27 d), even though the photosynthetically available radiation varies dramatically from season to season in the respective latitudes

(also because radiation experienced by phytoplankton cells dispersed in surface waters, is a function of the seasonally varying surface mixed layer depth). By chance, this could be the result of nonlinear forcing modulating a deficient nonlinear formulation of $PO_4$-limitation such that the model bias stays constant over a wide range of environmental conditions (here seasons). But this is unlikely. Looking into the seasonal bias of simulated iron concentrations (Fig. 29 d) we find that it varies substantially from season to season compared to the respective $PO_4$ variability (Fig. 27 d) – just as is expected when a deficient nonlinear

model formulation is exposed to substantial variations in driving environmental conditions.

The distribution of $PO_4$ and dissolved oxygen in the interior (c.f. Fig. 19)and Fig. 20) is consistent with an overly vivid biological pump that is not throttled enough by iron limitation: a prolonged overestimated biological production drives an overly high export of organic matter to depth (c.f. spurious $PO_4$ maximum around $400 \, m$ depth south of $60°$). Once in the thermocline this $PO_4$ is exported equatorwards and leaves the Southern Ocean (e.g. Marinov et al., 2006). Becoming part of

the meridional overturning circulation this excess $PO_4$ will return to the Southern Ocean at the surface. Because of insufficient iron limitation the excess will be used up and exported to depth into e.g. the Subantarctic Mode Water which leaves the Southern Ocean before it reaches higher latitudes. By this spurious mechanism the Southern Ocean looses $PO_4$ to the rest of the world's ocean.

The simulated patterns of dissolved oxygen bias in Fig. 20 is consistent with an overly efficient biological pump which strips the surface waters of $PO_4$ on their way from the world's ocean to higher latitudes, exports them to depth where they are remineralized and imprint a spurious oxygen deficit. Superimposed onto this, as a rough calculation suggests, is a negative bias of the order of $5\,mmol\,O_2\,m^{-3}$ caused by the temperature bias via the temperature-dependence of oxygen solubility. Note that a more comprehensive analysis is beyond the scope of this model description which is intended to describe the configuration rather than to evaluate it.

Fig. 28 summarizes the model's fidelity in reproducing observed patterns of biogeochemical: ranked against the very similar albeit coarser, non-eddying reference configuration used in Dietze et al. (2017) and UVic (here referring to the reference version used in Löptien and Dietze (2019) which is based on (Keller et al., 2012)) MOMSO appears to perform in a very similar manner.

## 3.1 Computational cost

Initial development, testing and spin-up of the physical configuration was carried out on two 32-core workstations, based on a 6320 AMD Opteron (Abu Dhabi) (8-core CPU, 2.80 GHz, 16MB L3 Cache, DDR3 1600) interconnected with a QDR InfiniBand. As of 2013 the system cost $\approx$ Euros 10.000,-. The fully coupled ocean-circulation biogeochemical model has been run at a supercomputing centre (project shk00027 at the *Norddeutscher Verbund für Hoch- und Höchstleistungsrechnen, HLRN*). Fig. 35 documents excellent parallel performance up to 500 tasks - a setting which optimized our throughput on the machine/queueing system. On 500 Intel Xeon CPU E5-2670 cores (Sandy Bridge) it takes 10 hours to simulate one year. Referring to pricing published by HLRN in 2018 this amounts to $\approx$ Euros 250,- per integrated model year. We use compiler settings that optimize for maximum computational efficiency. They violate computational reproducibility (c.f. intel fortran compiler option *-fp-model* and summary by Corden and Kreitzer (2018)). A description of the associated computational uncertainty is provided in Section 3.8 of Dietze et al. (2014) which use the same code (applied to a different domain) .

## 4 Research Questions

The purpose of this section is to outline research questions for which MOMSO may serve as a tool. MOMSO is (to our knowledge) unique in that it (1) features a high spatial resolution in the Southern Ocean which allows an eddying circulation in combination with (2) a biogeochemistry that is sufficiently equilibrated such that the remaining trend is small enough to allows for studying decadal signals in SO carbon uptake; and (3) it is a "free-forward-in-time-running" model in a sense that (projected or hind-casted) anomalies to air-sea fluxes can be applied easily (as opposed to in data-assimilated or inverse models where this can be difficult). In the following we sketch - very roughly - two ideas.

### 4.1 Wind-induced changes in Southern Ocean Carbon Uptake: parameterizing versus resolving the effects of eddies

Winds have been changing in the past decades over the Southern Ocean and they are anticipated to do so in the future. Lovenduski et al. (2013) estimates a linear increase at a rate of 14% in 50 years throughout the period 1958 to 2007. Dietze et al.

(2017) find in a coarse resolution model that such a trend in winds is associated to a decreasing trend in Southern Ocean natural carbon uptake between 4.2 and $5.4\,PgC/(1000\,yr^2)$ - depending on the choice of the underlying eddy-parameterization. An open question is how these estimates compare against a model configuration that is actually resolving the eddying circulation rather than just parameterizing it. The main problem preventing an answer to this question so far has been the computational expense associated with running the carbon dynamics of a realistic eddying coupled ocean-circulation biogeochemical model into equilibrium.

MOMSO is identical to the coarse resolution configurations used in Dietze et al. (2017), with the only difference being the higher spatial resolution and eddying circulation. Fig. 11, 15, 31 and 32 show that, the forcing of MOMSO with wind anomalies corresponding to a linear increase at a rate of 14% in 50 years like being estimated by Lovenduski et al. (2013), drives trends in sea surface temperatures, sea surface salinities and the Eularian overturning which are clearly distinguishable from the remaining model drift.

In line with the clearly detectable trends in physical properties in response to increasing winds Fig. 34 reveals that the associated trend in the oceanic uptake of natural carbon is also clearly detectable: the remaining drift in the MOMSO reference simulation is $1.0\,PgC/(1000\,yr^2)$ allowing for a clear detection of the effect of increasing winds in simulation WIND which yields a substantially larger decline of $5.0\,PgC/(1000\,yr^2)$.

Put into perspective: MOMSO yields a drift-corrected $5\,PgC/(1000\,yr^2 - 1\,PgC/(1000\,yr^2)) = 4\,PgC/(1000\,yr^2)$. Using the same code framework and exact-same biogeochemical module (but much coarser spatial resolution) Dietze et al. (2017) find between 4.2 and $5.4\,PgC/(1000\,yr^2)$ depending on the underlying eddy-parameterization. So this is an indication that contemporary eddy-parameterizations may provide results very similar to high-resolution eddying configurations.

Further investigations, which are beyond the scope of this sketch of a potential application of MOMSO must include an in-depth analysis of "eddy-compensation" which is capable of canceling changes in wind-driven overturning (c.f. Marshall and Radko, 2003; Viebahn and Eden, 2010; Abernathey et al., 2011, 2016; Hallberg and Gnanadesikan, 2006; Thompson et al., 2014; Tamsitt et al., 2017).

### 4.2 Wind-induced anomalies in Antarctic Continental Shelf Bottom Water temperatures

The realistic seabed temperatures (shown in Fig. 13), in combination with its biogeochemical module, provide a starting point which - with further development - may be suited to explore the following feedback loop with MOMSO: atmospherically-driven changes in ocean-circulation drive additional heat supply which lead to basal melting. The buoyant lens of meltwater may, for one, suppress the AABW-formation (Williams et al., 2016). This, in turn, effects oceanic carbon sequestration. Second, the meltwater carries bioavailable iron to the Southern Ocean - which affects oceanic primary productivity and the associated export of organic carbon to depth (Grotti et al., 2005; Lannuzel et al., 2008, 2010; Raiswell et al., 2008; Smith et al., 2007; Smith and Nelson., 1986; van der Merwe et al., 2009).

Fig. 33 shows that the simulated ASBW changes its temperature in response to a linear wind increase a rate of 14% in 50 years like being estimated by Lovenduski et al. (2013). The magnitude of ASBW temperature is distinguishable from the persisting model drift in the reference simulation (c.f. panel a and b in Fig. 33) and simulated magnitudes are similar to recent

observational estimates (c.f. panel b and c in Fig. 33). Schmidtko et al. (2014). The sign, however, is reversed which suggests that observed changes of ASWB temperatures can not be attributed to the recent increases in wind-strength alone. In any case, Fig. 33 suggests that the equilibration of the model has set in sufficiently so that model responses to decadal forcing variability are detectable.

## 5   Summary and conclusions

We set out to develop a near-global coupled ocean-circulation biogeochemical model which explicitly resolves – in contrast to relying on respective parameterizations – effects of mesoscale eddy-dynamics on the uptake of carbon in the Southern Ocean. The setup is dubbed MOMSO with MOM referring to the underlying framework (GFDL's modular ocean model version MOM4p1) and SO referring to the Southern Ocean. We use the biogeochemical/carbon module BLING developed by Galbraith et al. (2010).

Overall, we find in a climatological simulation (REF) an eddying surface circulation that is in reasonable agreement with observations from space. Further, simulated temperatures and sea surface salinity show a close agreement with observations. Sea-ice cover is biased high but is still in the range of observed values during particularly cold winters. A remaining caveat is a low bias in the Drake Passage ($99\,Sv$ compared to observational estimates ranging from $110$ to $170\,Sv$). This may be associated to locally enhanced viscosity over rough topography at the exit of the Drake Passage (c.f. 2) which has been necessitated by numerical stability. (Whether this is associated to our very low choice of the Smagorinsky isotropic viscosity and/or the lack of biharmonic viscosity/diffusion as suggested by review 1 still warrants investigation.)

The simulated biogeochemistry is also biased. Surface $PO_4$ concentrations are too low in the Southern Ocean which is indicative of a deficient formulation of the limitation of phytoplankton growth. Seasonally varying biases in simulated surface iron concentrations suggest that the problem is associated to an, up-to-date, uncomprehensive quantitative understanding of iron-dynamics. The model performance with respect to biogeochemistry is similar to what is state-of-the-art in coarse resolution models.

MOMSO is a step forward in the field of eddying coupled ocean-circulation biogeochemical carbon modeling in that it allows to investigate the effects of decadal changes in atmospheric boundary conditions on the oceanic carbon uptake with an eddy-resolving model that features a realistic geometry. Previous attempts have been hindered by the computational cost that is associated to run simulations into a semi-equilibrated state which features trends considerably lower than the climate signals (as effected by prescribed anomalies in boundary conditions) under investigation. To this end MOMSO benefitted from: (1) an ever increasing ease of access to computing power that is associated to Moore's law and (2) the fortunate coincidence that the spun-up coarse resolution restart from Dietze et al. (2017) was close enough to the equilibrated high-resolution state of MOMSO.

In this model-description paper we showcased that the remaining drift in MOMSO's Southern Ocean carbon uptake is substantially lower than changes driven by typical decadal variability of atmospheric variations. We illustrate that the respective sensitivity experiment is suitable for a comparison to coarser-resolution model versions and present a first impression in Fig. 34.

Similarly we find that the drift in simulated temperatures of the Antarctic Continental Shelf Bottom Water is small enough to allow to link atmospheric decadal variability to oceanic temperature variations at the boundary to ice shelves around Antarctica.

We illustrate that the respective sensitivity experiment is suitable for a comparison to coarser resolution model versions and present a first impression in Fig. 25.

5     On a final note, the computational cost associated with integrating MOMSO is substantial. With present-day hardware full spin-ups are in the 5-figure Euro range so that extensive tuning of the biogeochemical model, which still features some substantial misfits to observations, is uneconomical. Starting from the already spun-up state presented here, however, is in the reasonable range of 2500,- Euros per decade. The corresponding energy needs (operation only) are of the order of $1000 kWh$ per decade. Using a carbon intensity of $450 gCO_2/kWh$ (e.g., Moro and Lonza, 2018) this yields an emission of around 0.5 10  tons of $CO_2$ per simulated decade.

*Code and data availability.*   The circulation model code MOM4p1 is distributed by NOAA's Geophysical Fluid Dynamics Laboratory (http://www.gfdl.noaa.gov/fms). We use the original code without applying any changes to it apart from very minor changes ($\approx$ 10 lines of code) which levitate sea ice (c.f., Sect. 2.3) and increase local viscosity (c.f., Sect. 2.2 and 2.5). The respective code, initial conditions, forcing, namelists, and model output is accessible via http://data.geomar.de/thredds/catalog/open_access/dietze_et_al_2018_gmd/catalog.html.

15  *Author contributions.*   H. Dietze and U. Löptien have been involved in setting up and running the model configuration. All three authors contributed to the interpretation of model results, to outlining and writing of the paper.

*Competing interests.*   The authors declare that they have no conflict of interest.

*Acknowledgements.*   Eric Galbraith, contributor to the MOM (www.gfdl.noaa.gov/mom-ocean-model/) community and developer of BLING (www.sites.google.com/site/blingmodel/), shared his biogeochemical model code with us. We are grateful to him and the rest of the MOM 20  community! Discussions with Ivy Frenger are appreciated. All authors acknowledge support by the Helmholtz Association of German Research Centres (HGF - grant no. ZT-I-0010 Reduced Complexity Models). U. Löptien and H. Dietze acknowledge additionally support by Deutsche Forschungsgemeinschaft SPP 1158, Glacial/Interglacial Hydrographic Structures and Nutrient Utilization in the Pacific Southern Ocean - Data and Modeling Approach (grant no. SCHN 762/5-1) and the BMBF-funded project PalMod 4.1. The authors acknowledge the North-German Supercomputing Alliance (HLRN) for providing HPC resources that have substantially contributed to the research results 25  reported in this paper. The authors acknowledge computing time on the joint GEOMAR/University Kiel NEC HPC-Linux-Cluster. Initial configuration development and testing has been carried out on the FB2/BM compute clusters weil.geomar.de and wafa.geomar.de located at the GEOMAR Helmholtz Centre for Ocean Research, west shore campus, Kiel, Germany. We acknowledge help from Kai Grunau for cluster maintenance and IT-services. The altimeter products were produced by Ssalto/Duacs and distributed by Aviso, with support from Cnes

(http://www.aviso.altimetry.fr/duacs/). SMOS ocean surface salinity was distributed in netCDF format by the Integrated Climate Data Center (ICDC, http://icdc.cen.uni-hamburg.de) University of Hamburg, Hamburg, Germany. Some of the data used in this study was collected and made freely available by the International Argo Program and the national programs that contribute to it. (http://www.argo.ucsd.edu, http://argo.jcommops.org). The Argo Program is part of the Global Ocean Observing System. We are grateful to all observational programs pursuing an open data policy. Constructive comments by two knowledgable reviewers helped to improve this manuscript substantially.

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

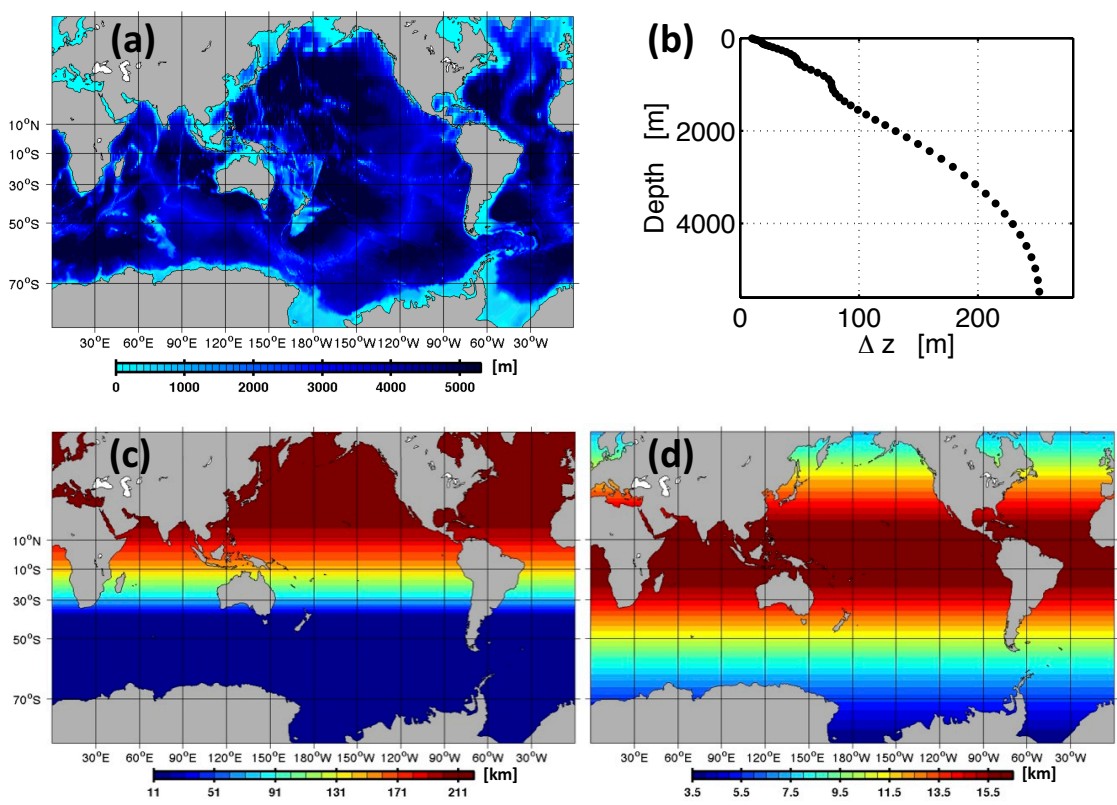

**Figure 1.** MOMSO model domain and spatial (finite differences) discretization. Panel **(a)** shows the model bathymetry. Panel **(b)**, **(c)** and **(d)** show the vertical, meridional and zonal resolution, respectively. In total, there are $2400 \times 482 \times 55$ grid boxes in zonal, meridional and vertical direction, respectively.

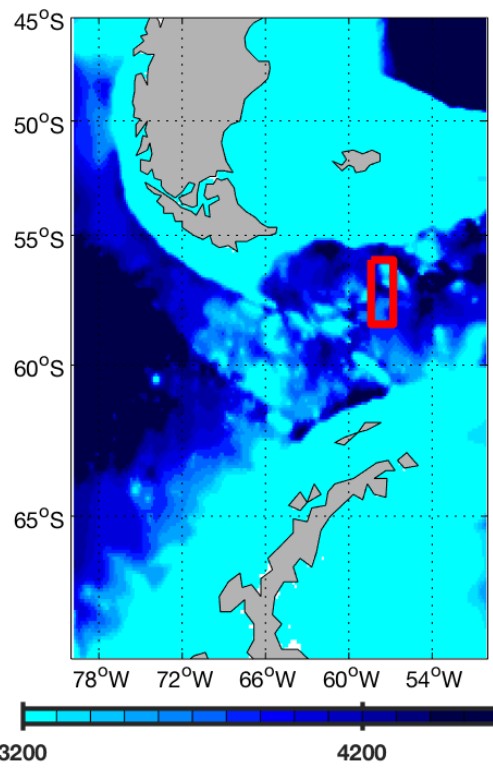

**Figure 2.** Closeup of MOMSO bathymetry in the Drake Passage. The color denotes depths in meters. The red rectangle denotes the area where, from 3250 m down to the bottom, an additional Laplacian horizontal viscosity of $1000\,m^2/s$ is applied for stability reasons.

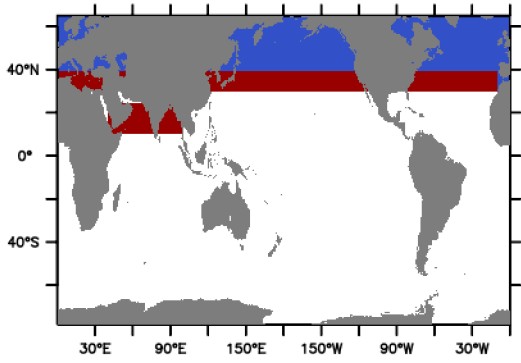

**Figure 3.** MOMSO domains where temperature and salinity are restored to observed climatological values (Locarnini et al., 2010; Antonov et al., 2009, respectively) throughout the water column. The red and blue patch denote e-folding restoring timescales of 10 and 5 years, respectively.

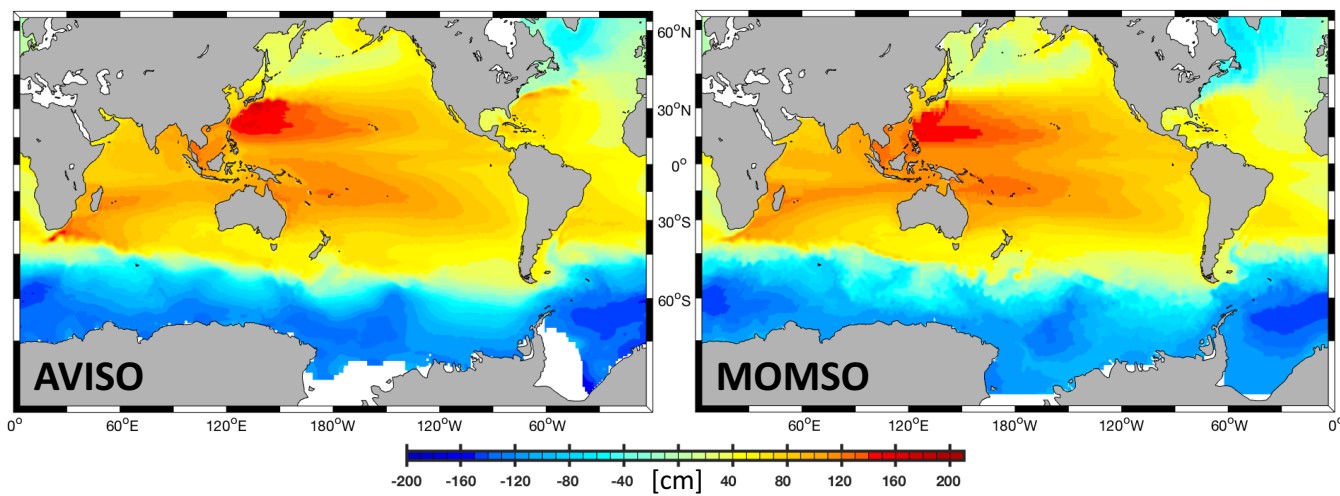

**Figure 4.** Climatological mean sea surface height in units $cm$. The left panel shows a 1993 to 1998 climatological average observed from space (MADT AVISO data). The right panel shows a 6-year average (nominal years 1993 to 1998) simulated with the reference simulation. White patches in the left and right panel indicate missing data and (spurious) model land mask, respectively.

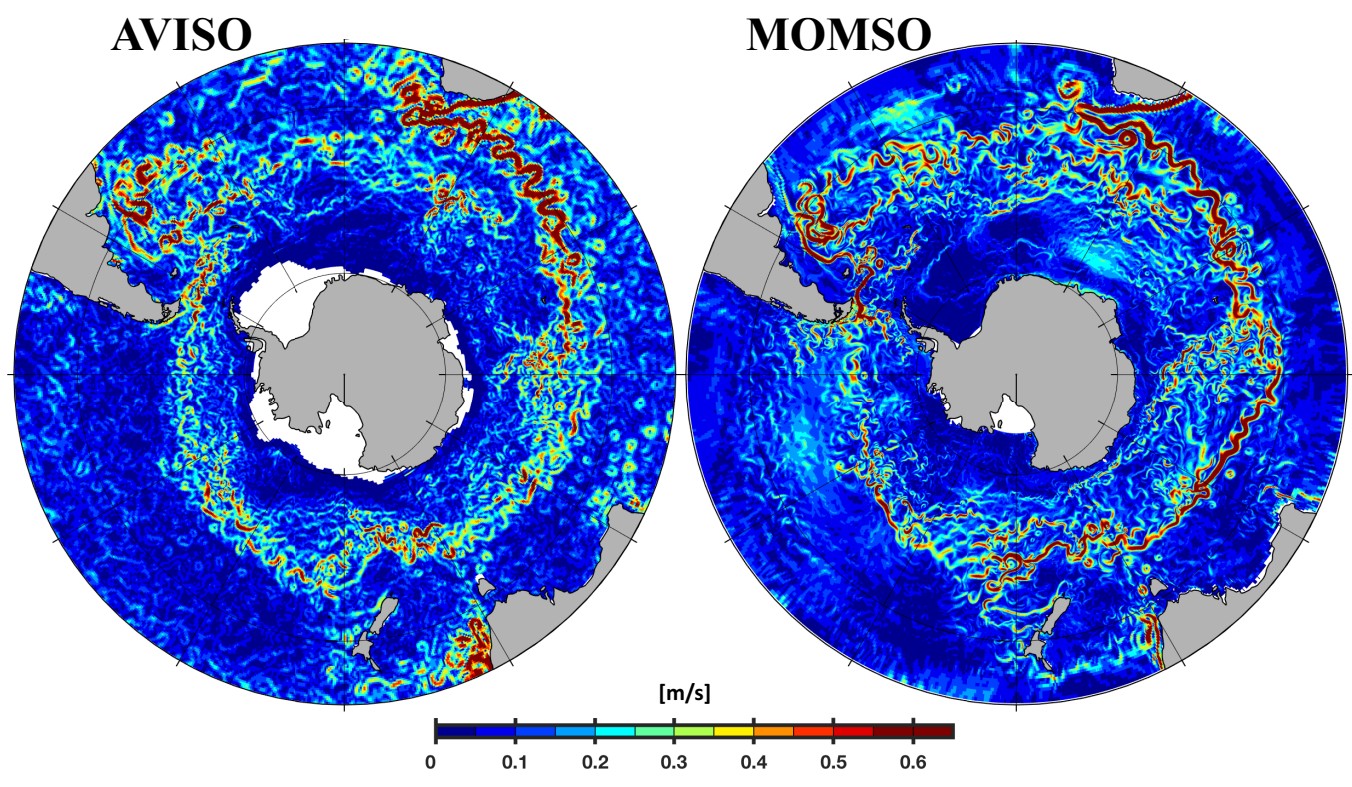

**Figure 5.** Magnitude of surface velocities. The left and right panel show typical snapshots as observed from space (AVISO, 3rd of May 1995) and as simulated (MOMSO, 3rd of May of nominal year 2020), respectively.

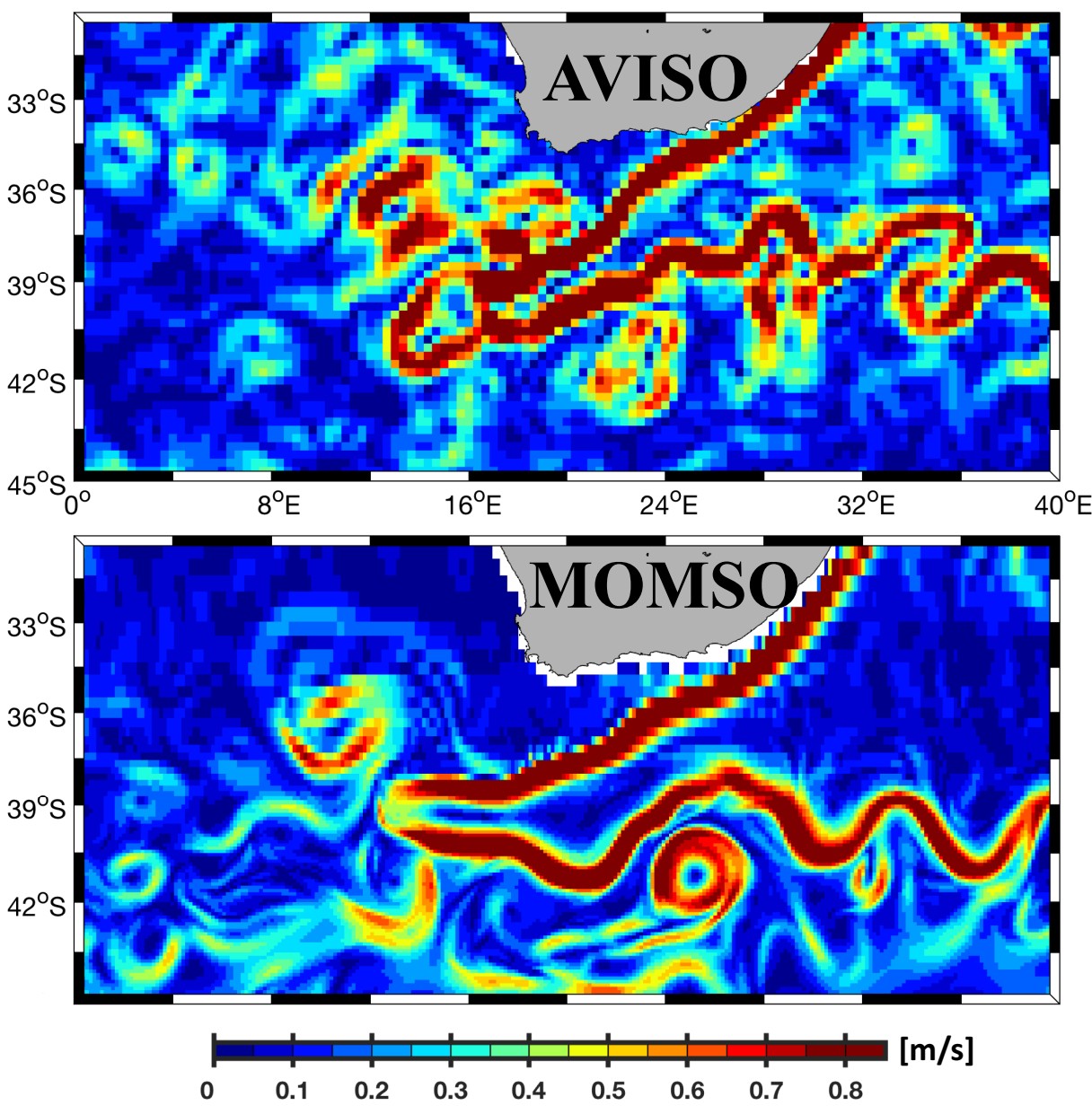

**Figure 6.** Magnitude of surface velocities. The left and right panel show typical snapshots as observed from space (AVISO, 3rd of May 1995) and as simulated (MOMSO, 3rd of May of nominal year 2020), respectively. This is a closeup from Fig. 5, focussing on those latitudes where the meridional resolution transitions from eddy-permitting in the South to eddy-prohibiting in the North.

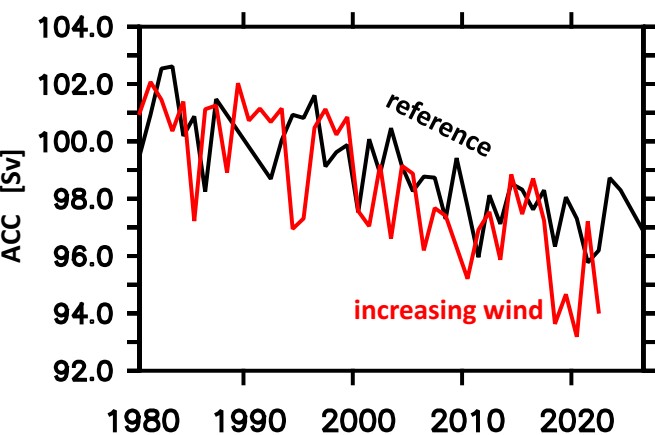

**Figure 7.** Simulated volume transport of the Antarctic Circumpolar Current through Drake Passage in units $10^6 \, m^3 \, s^{-1}$. The black (red) line refers to the reference (increasing-wind) simulation.

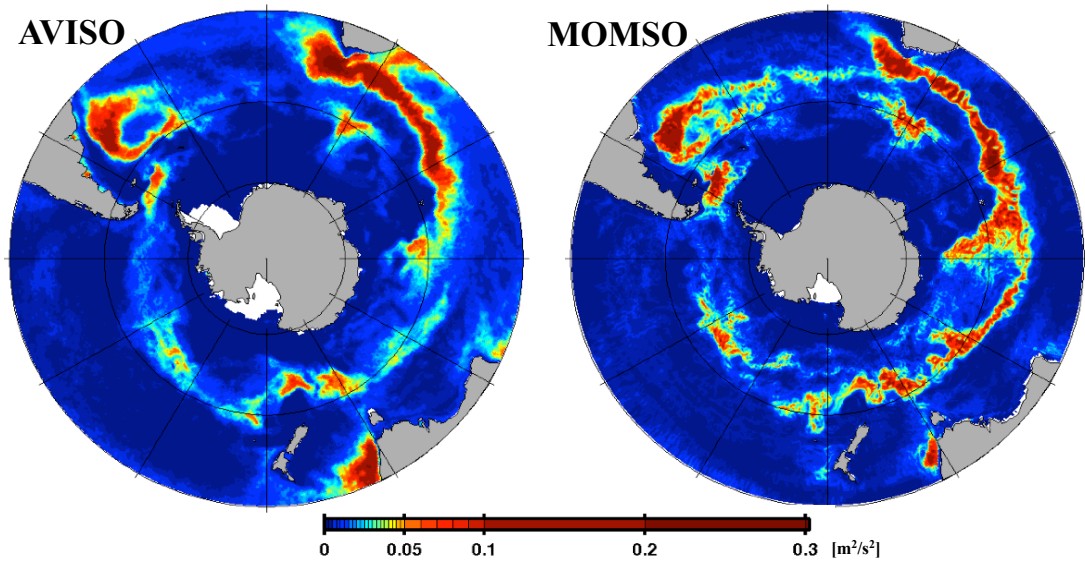

**Figure 8.** Eddy kinetic energy. The left panel is calculated from 1993-1998 satellite altimetry (MSLA AVISO data) observations. The right panel corresponds to a 6-year average (nominal years 1993 to 1998) calculated from daily-averaged surface velocities from the reference simulation. White patches in the left and right panel indicate missing data and (spurious) model land mask, respectively.

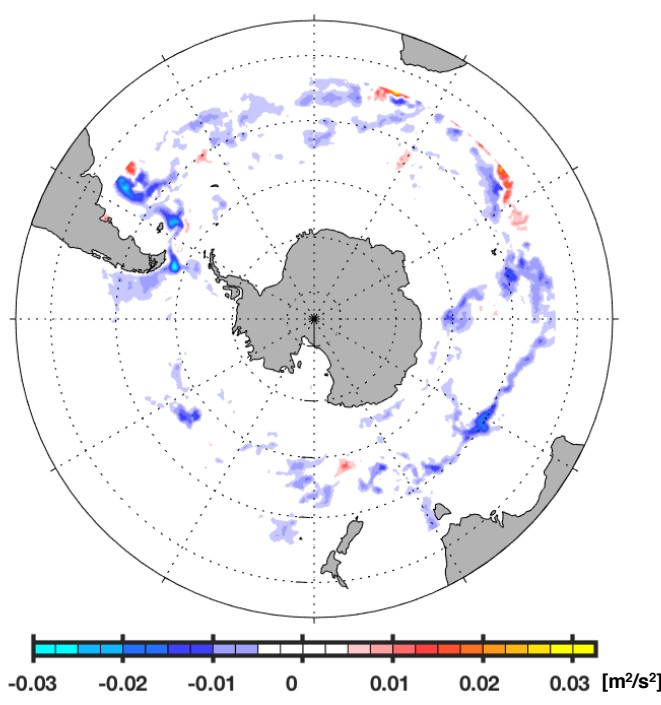

**Figure 9.** Difference between observed and simulated eddy kinetic energy. Blue colors denote regions where the observed levels calculated from 1993-1998 satellite altimetry (MSLA AVISO data) observations are less than simulated levels. The region outside the high-resolution nest (north of $40°$S) is masked by a white patch.

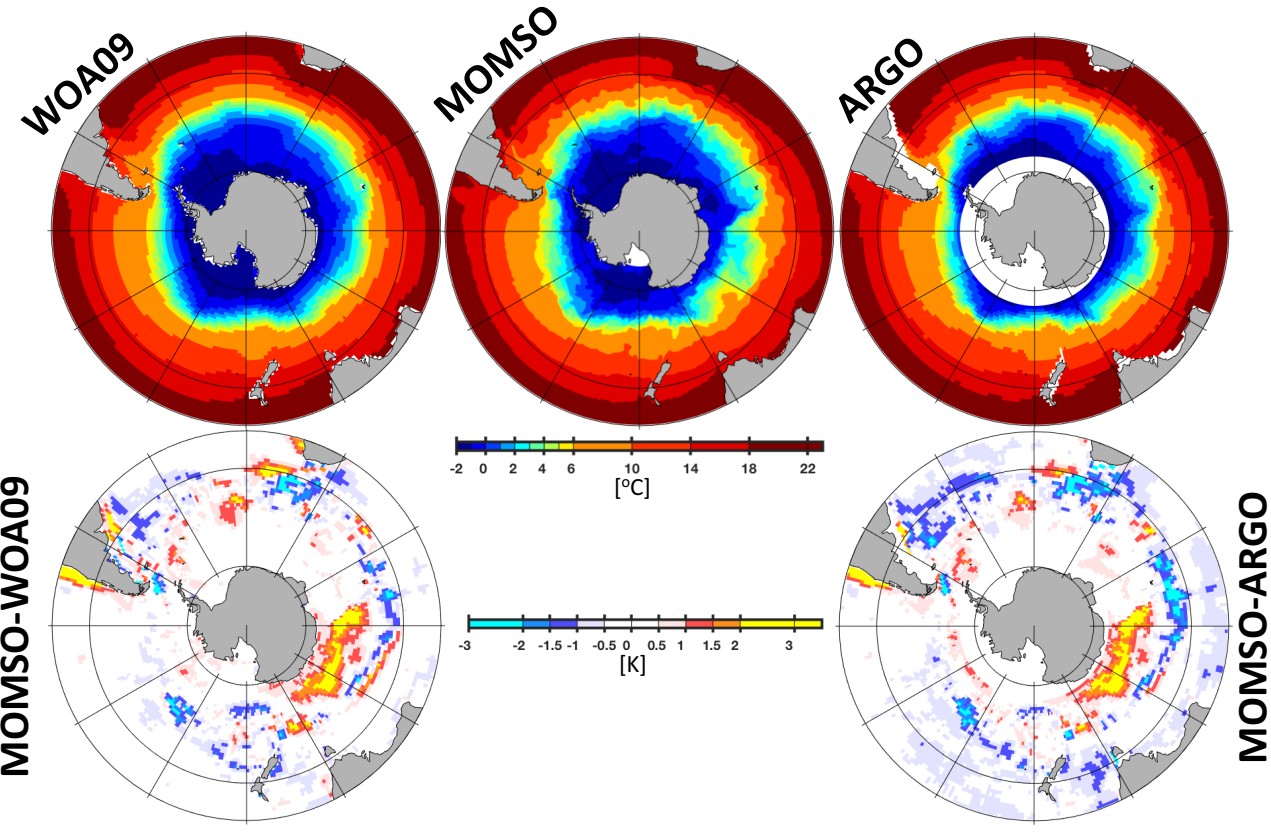

**Figure 10.** Climatological mean sea surface temperature. WOA09 and ARGO (2004-2017 period) refer to observations compiled by Locarnini et al. (2010), and Roemmich and Gilson (2009), respectively. MOMSO refers to an average over the nominal 1993-1998 period of the reference simulation. The upper (lower) panels show sea surface temperature (differences).

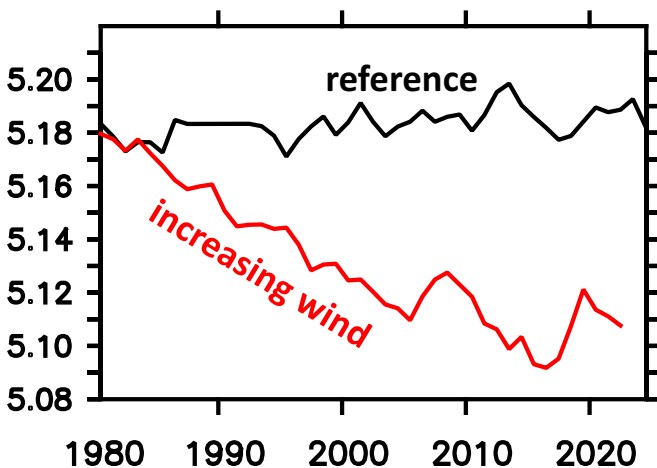

**Figure 11.** Simulated sea surface temperature averaged over the Southern Ocean (i.e. south of $40°$S). The black (red) line refers to the reference (increasing-wind) simulation.

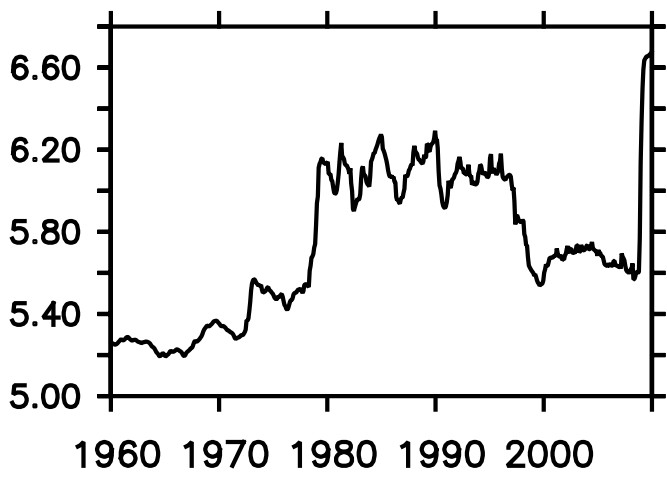

**Figure 12.** Observational estimate of annual mean sea surface temperatures south of $40°$S calculated from HadISST (Rayner et al., 2003).

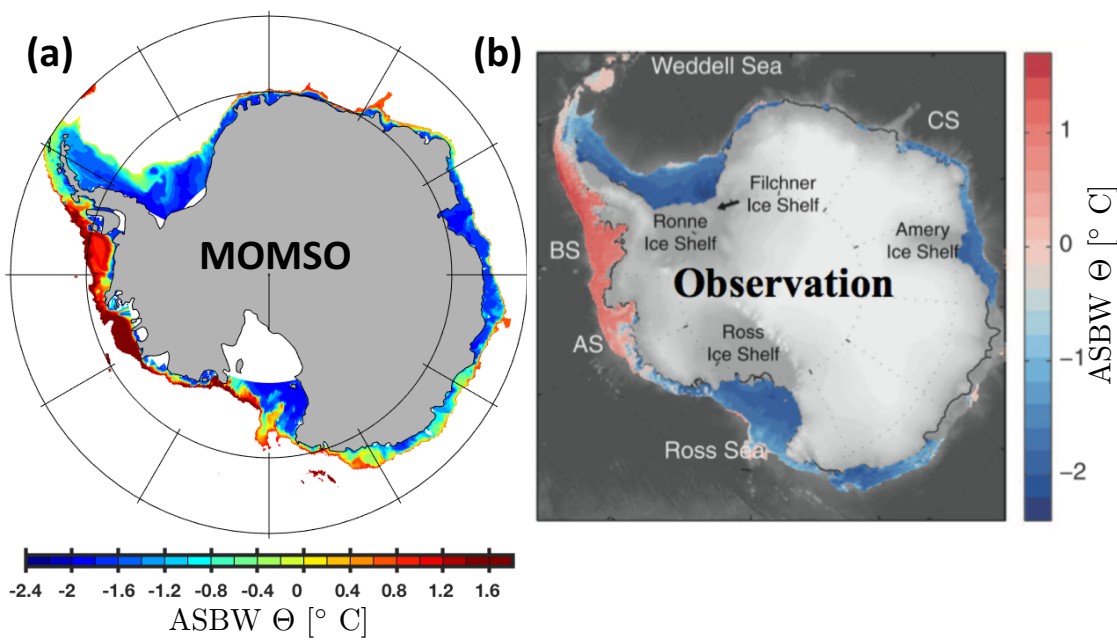

**Figure 13.** Temperature of Antarctic Continental Shelf Bottom Water (ASBW) at the seabed for depths shallower than 1500 m. Panel **(a)** and **(b)** refer to the simulated 1993 to 1998 climatology and observations compiled by Schmidtko et al. (2014), respectively.

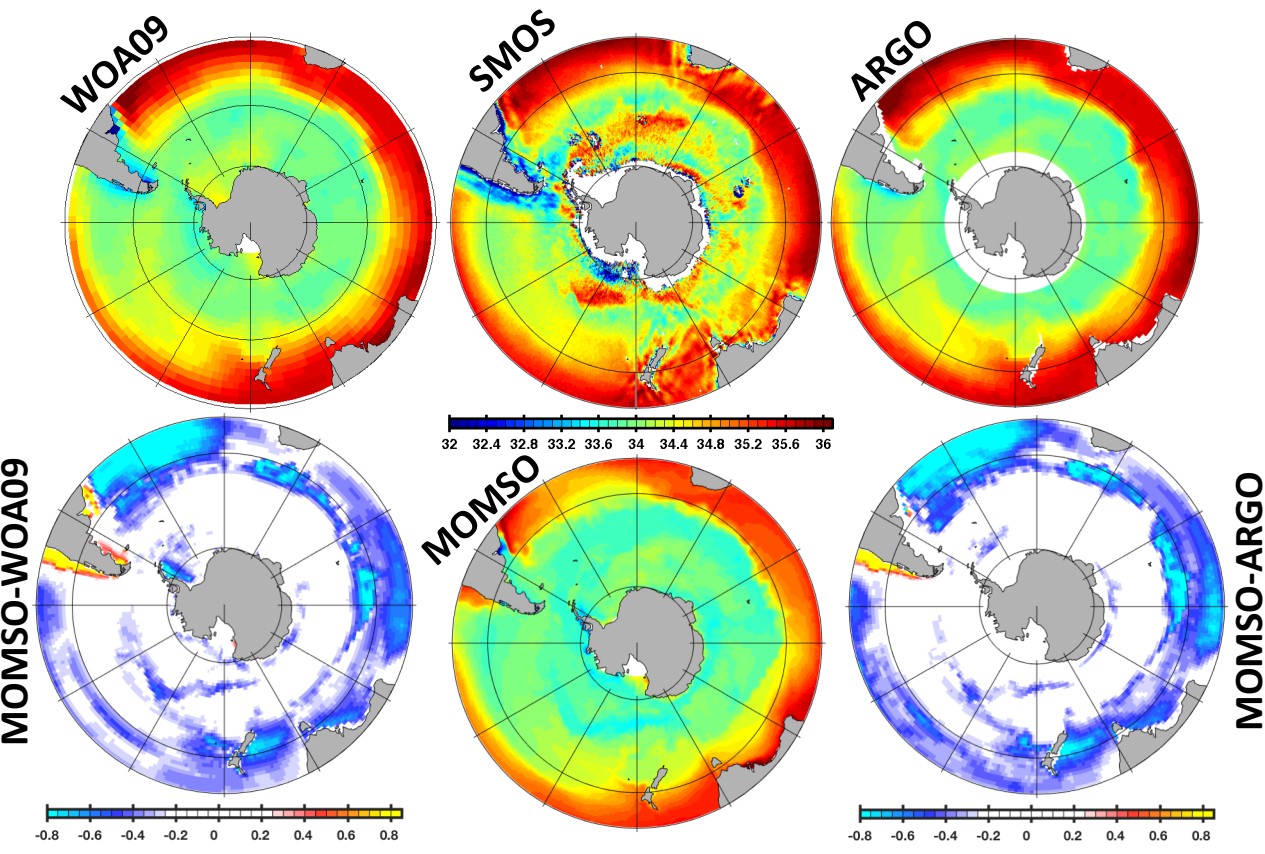

**Figure 14.** Climatological mean sea surface salinity in units $PSU$. WOA09, ARGO (2004-2017 period) and SMOS in the upper panels refer to observations compiled by Antonov et al. (2009), Roemmich and Gilson (2009) and Köhler et al. (2015), respectively. MOMSO in the lower panel refers to an average over the nominal 1993-1998 period of the reference simulation. MOMSO-WOA09 and MOMSO-ARGO refer to sea surface salinity differences between the reference simulation and respective observations.

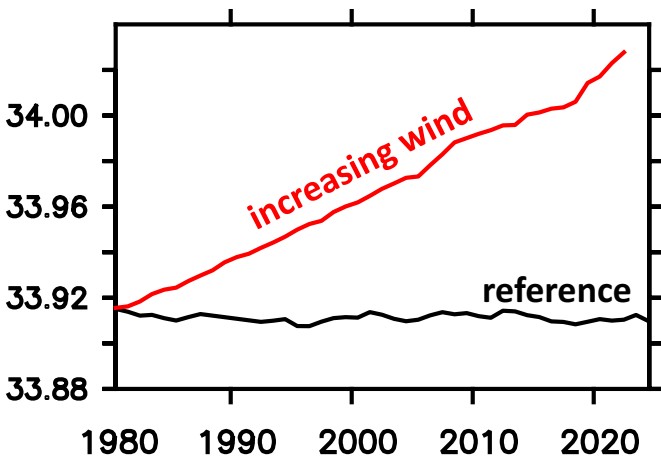

**Figure 15.** Simulated sea surface salinity averaged over the Southern Ocean (i.e. south of $40°$S) for the nominal years 1980 – 2024. The black (red) line refers to the reference (increasing-wind) simulation.

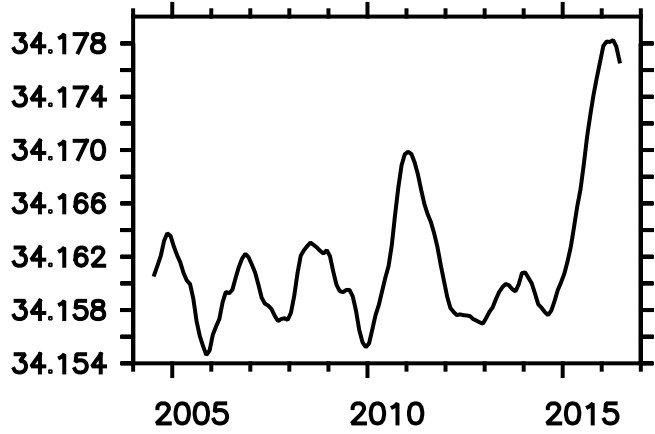

**Figure 16.** Observed sea surface salinity averaged over the Southern Ocean (i.e. south of $40°$S) based on ARGO data (Roemmich and Gilson, 2009).

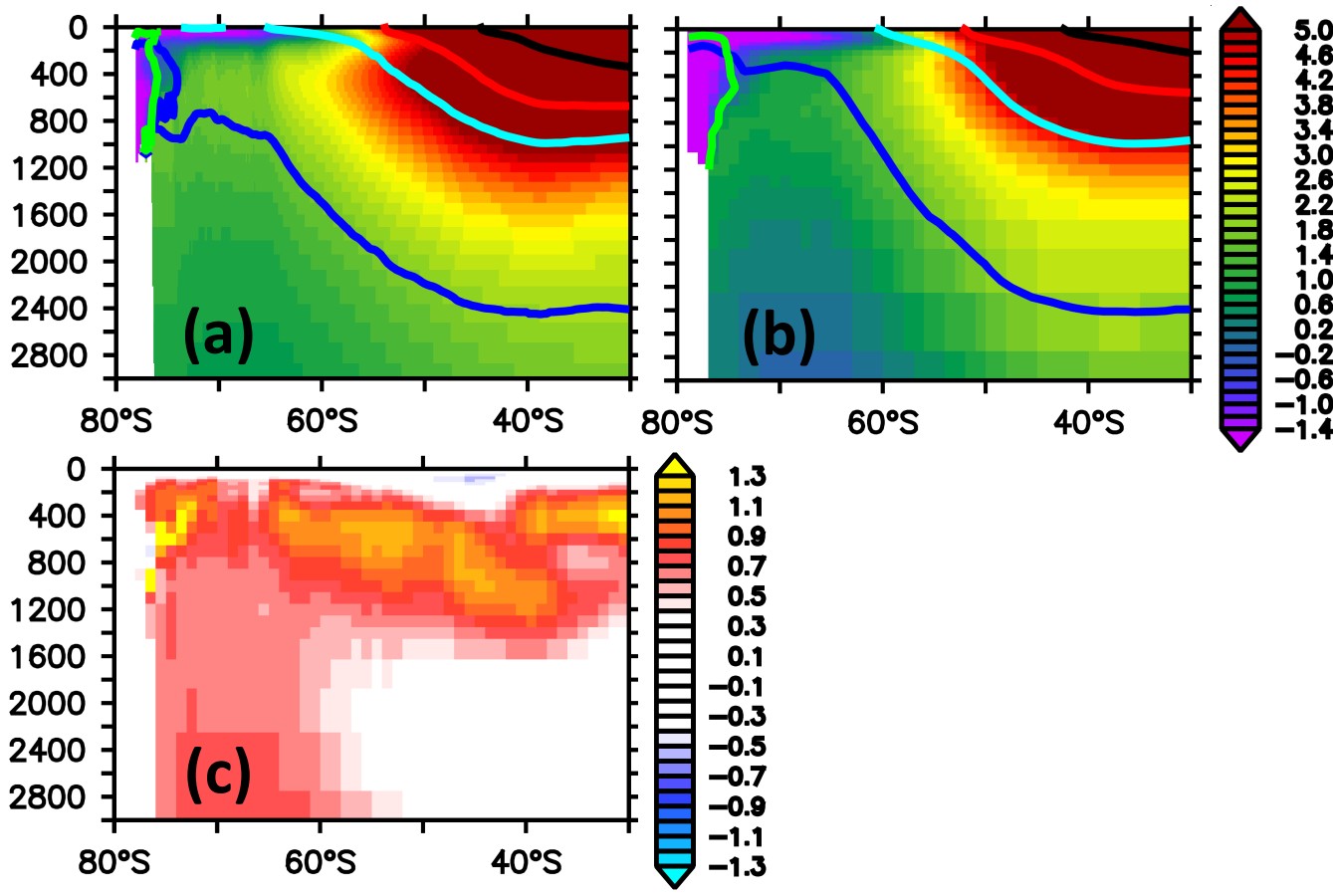

**Figure 17.** Zonally averaged, meridional section of temperature in units $^{\circ}C$. (a) refers to simulated concentrations averaged over the nominal 1993-1998 period of the reference simulation. (b) refers to observed climatological values (WOA09). (c) refers to the difference between simulated and observed values, with blue colors denoting simulated temperatures that are biased low. The thick black, red, cyan, blue and green contours refer to mean densities (Sallée et al., 2013, their table 2) of subtropical water, mode water, intermediate water, circumpolar deep water and bottom water, respectively.

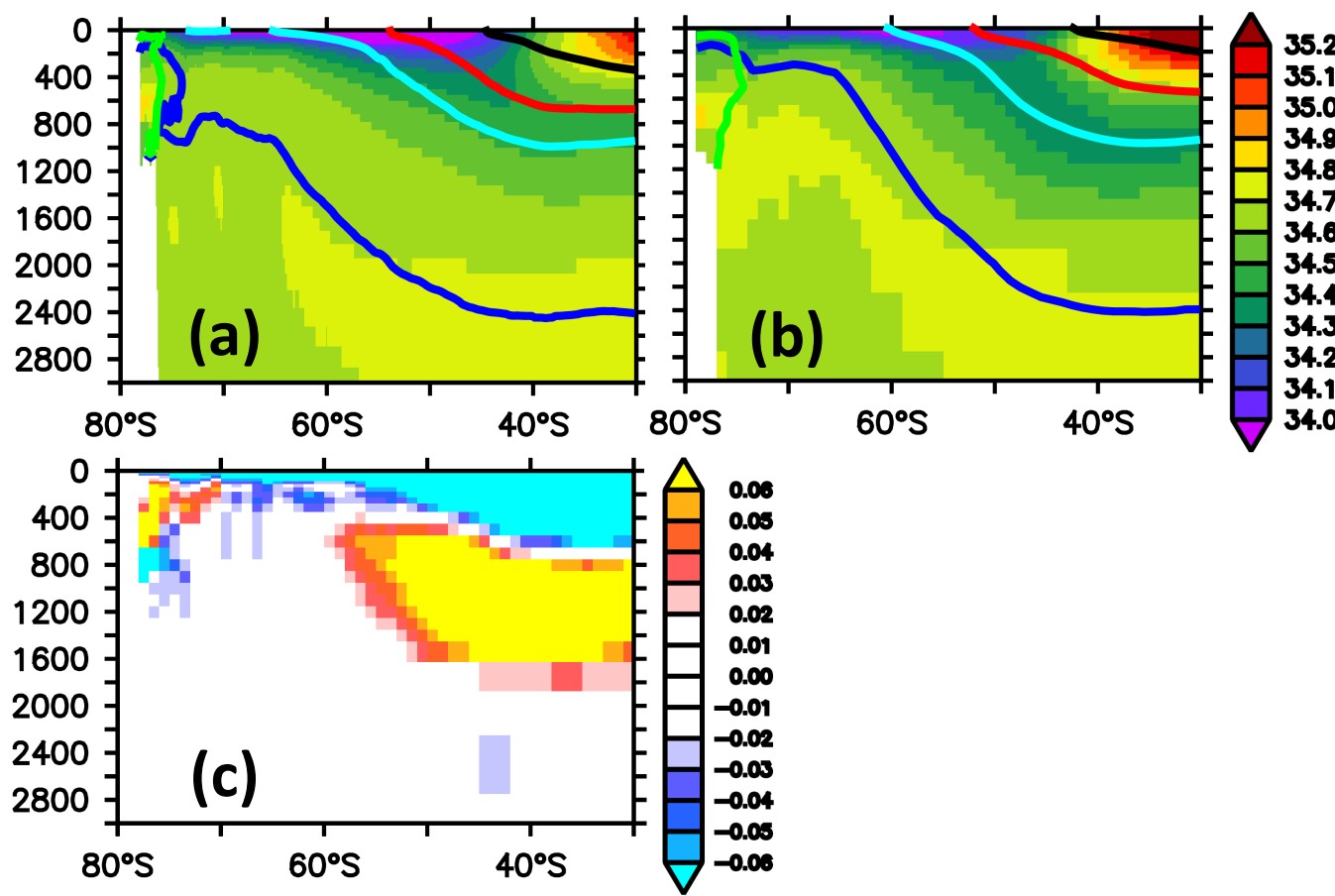

**Figure 18.** Zonally averaged, meridional section of salinity. (a) refers to simulated salinities averaged over the nominal 1993-1998 period of the reference simulation. (b) refers to observed climatological salinities (WOA09). (c) refers to the difference between simulated and observed values, with blue colors denoting simulates salinities that are biased low. The thick black, red, cyan, blue and green contours refer to mean densities (Sallée et al., 2013, their table 2) of subtropical water, mode water, intermediate water, circumpolar deep water and bottom water, respectively.

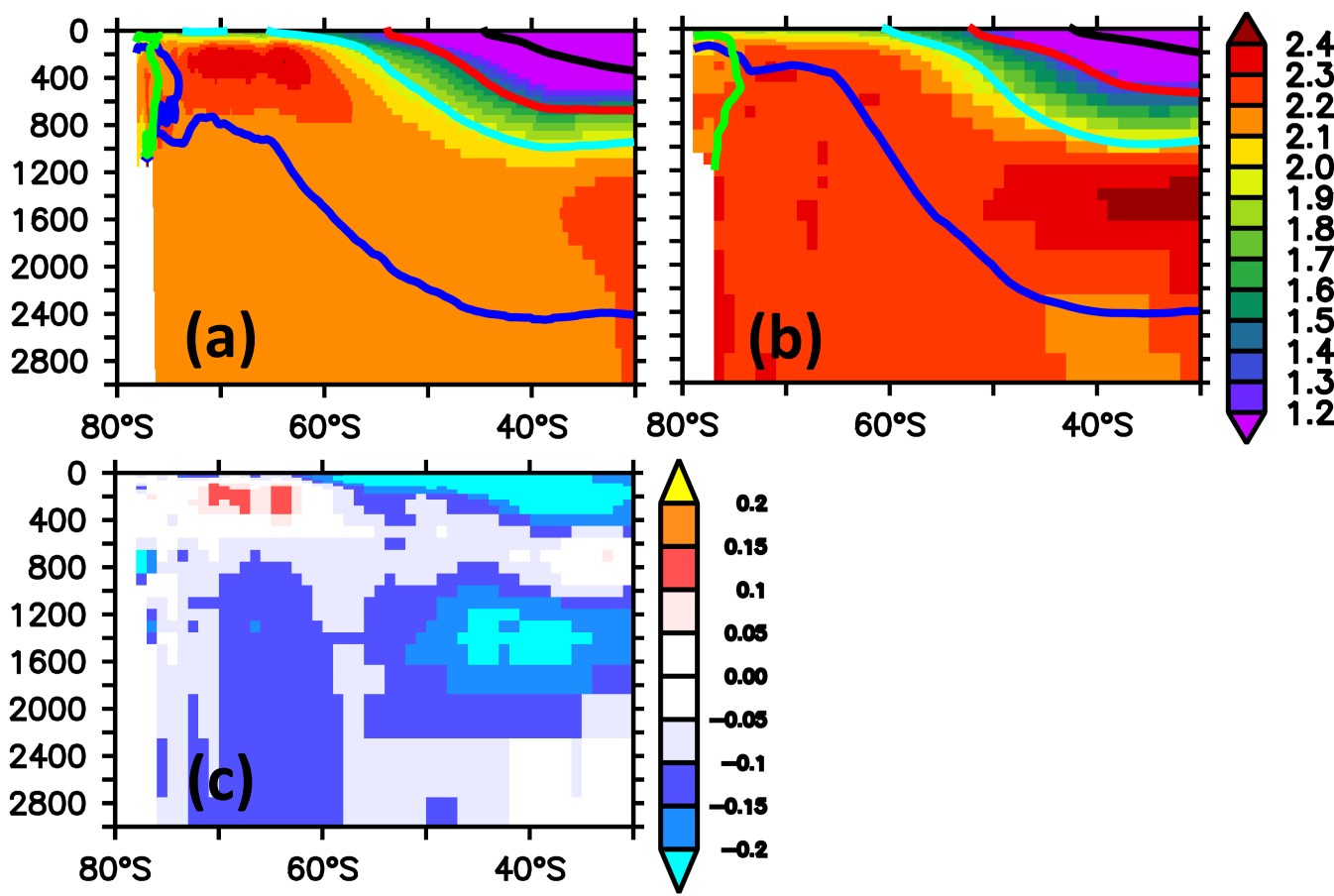

**Figure 19.** Zonally averaged, meridional section of phosphate concentration in units $mmol\, P/m^3$. (a) refers to simulated concentrations averaged over the nominal 1993-1998 period of the reference simulation. (b) refers to observed climatological concentrations (WOA09). (c) refers to the difference between simulated and observed concentratio ns, with blue colors denoting modeled concentrations that are biased low. The thick black, red, cyan, blue and green contours refer to mean densities (Sallée et al., 2013, their table 2) of subtropical water, mode water, intermediate water, circumpolar deep water and bottom water, respectively.

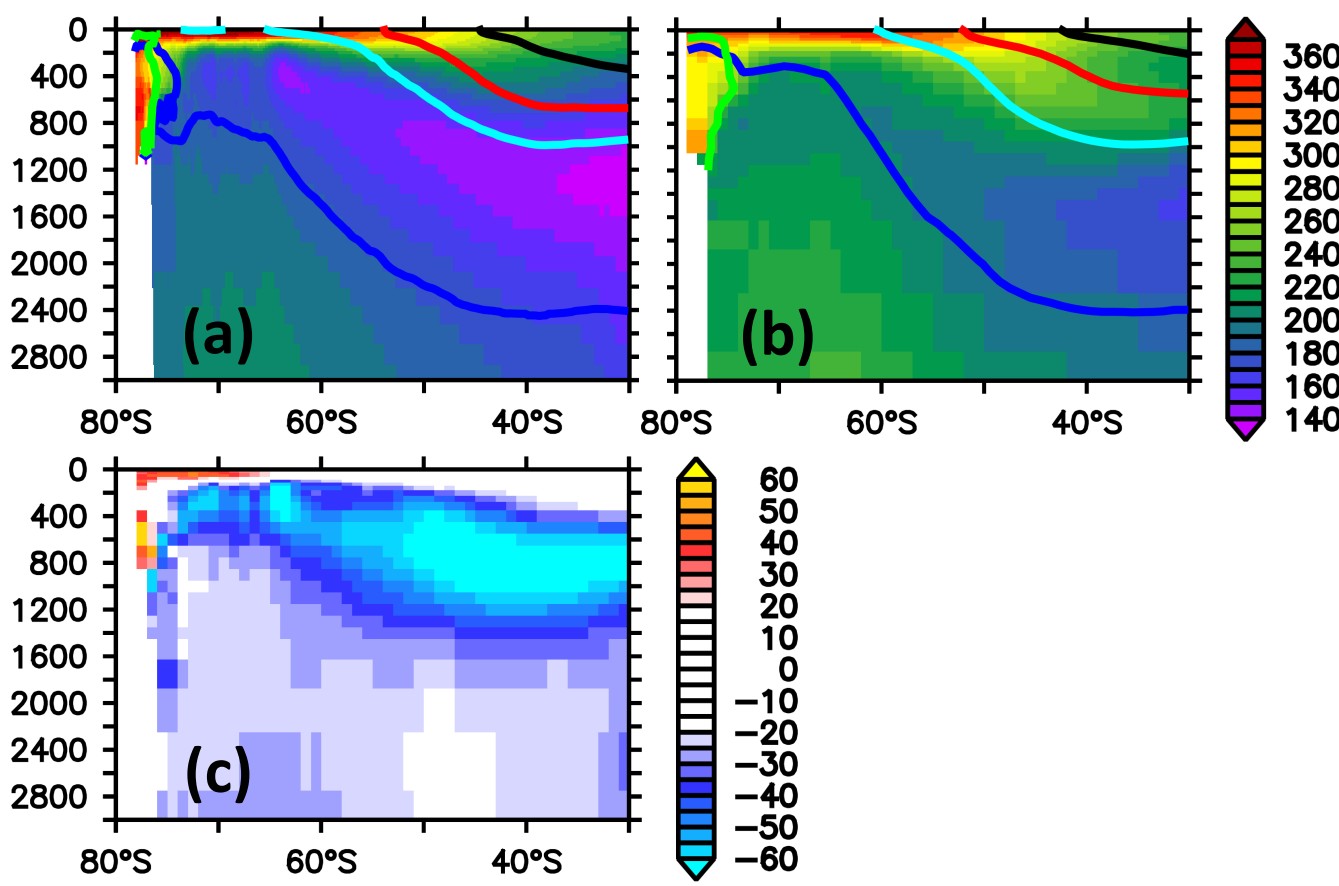

**Figure 20.** Zonally averaged, meridional section of dissolved oxygen concentration in units $mmol/m^3$. (a) refers to simulated concentrations averaged over the nominal 1993-1998 period of the reference simulation. (b) refers to observed climatological concentrations (WOA09). (c) refers to the difference between simulated and observed concentrations, with blue colors denoting modeled concentrations that are biased low. The thick black, red, cyan, blue and green contours refer to mean densities (Sallée et al., 2013, their table 2) of subtropical water, mode water, intermediate water, circumpolar deep water and bottom water, respectively.

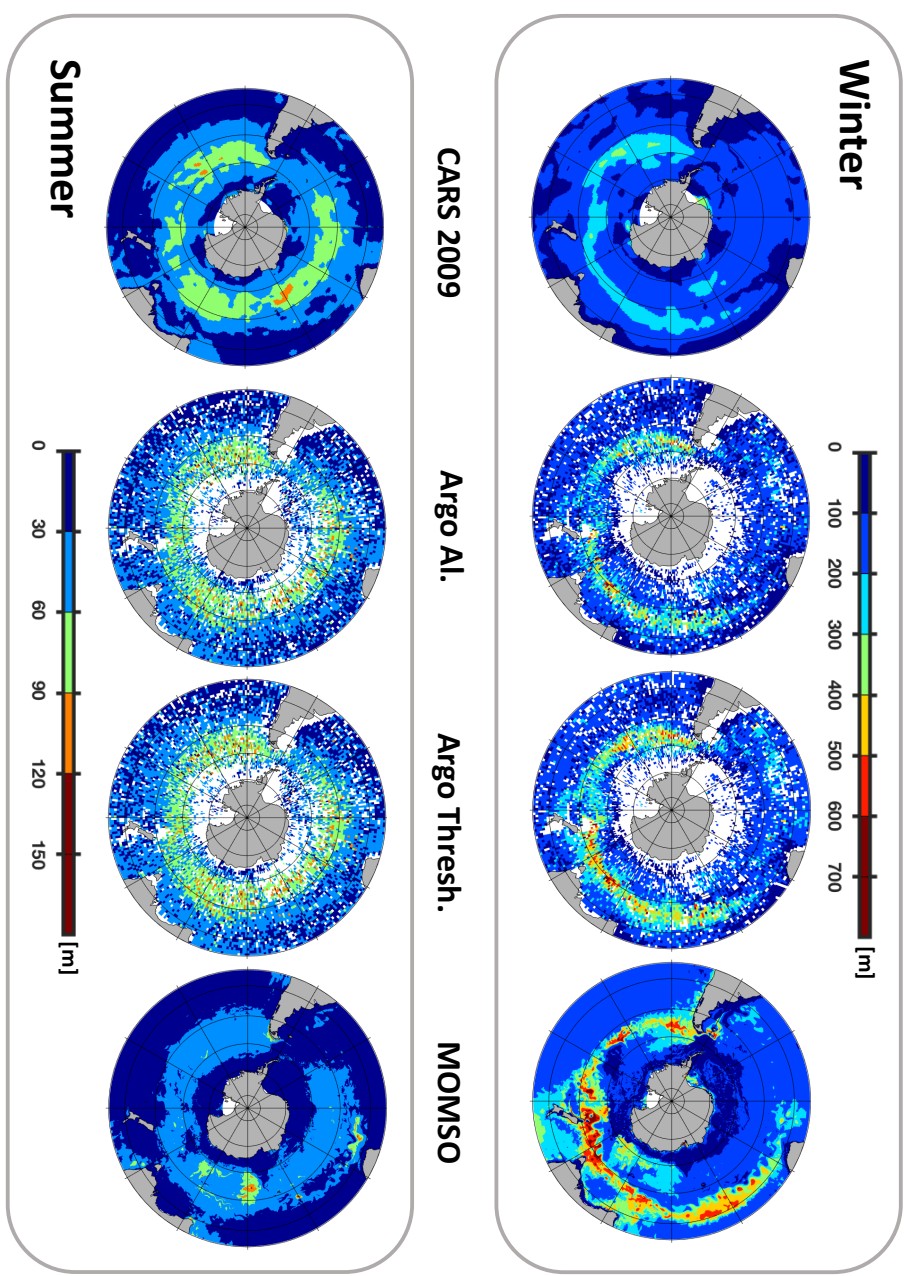

**Figure 21.** Surface Mixed layer depth in units $m$. The upper (lower) line refers to Austral winter (summer). The columns refer to (observed) climatology CARS 2009 based on Condie and Dunn (2006), *Argo Al.* and *Argo Thresh.* computed from (observed) profiles of Argo floats using two different methods by Holte et al. (2017), and our simulation MOMSO. White patches in *Argo Al.* and *Argo Thresh.* denote missing data.

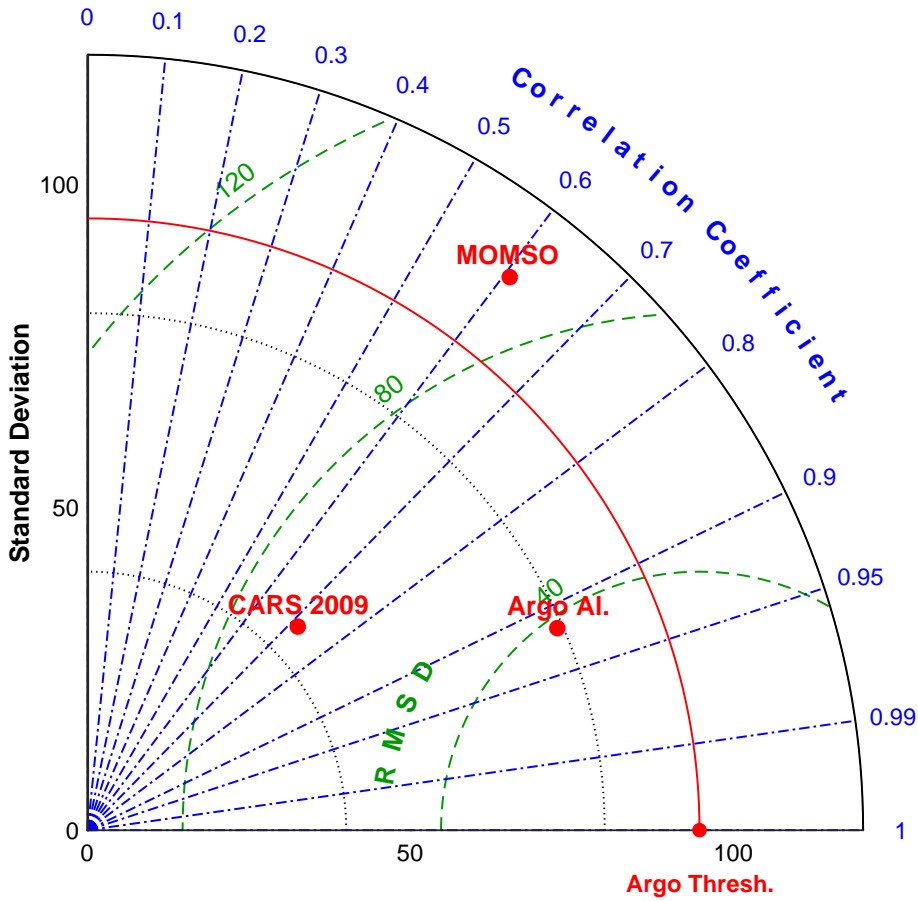

**Figure 22.** Taylor diagram of surface mixed layer depth in Austral winter referenced to *Argo Thresh.* observational data compiled by Holte et al. (2017). The units for the standard deviation and root-mean-square deviations are $m$. *Argo Al.* refers also to observational data compiled by Holte et al. (2017) but based on a different algorithm. CARS 2009 is a climatology based on observations compiled by Condie and Dunn (2006) and MOMSO refers to the high-resolution reference simulation presented in this study.

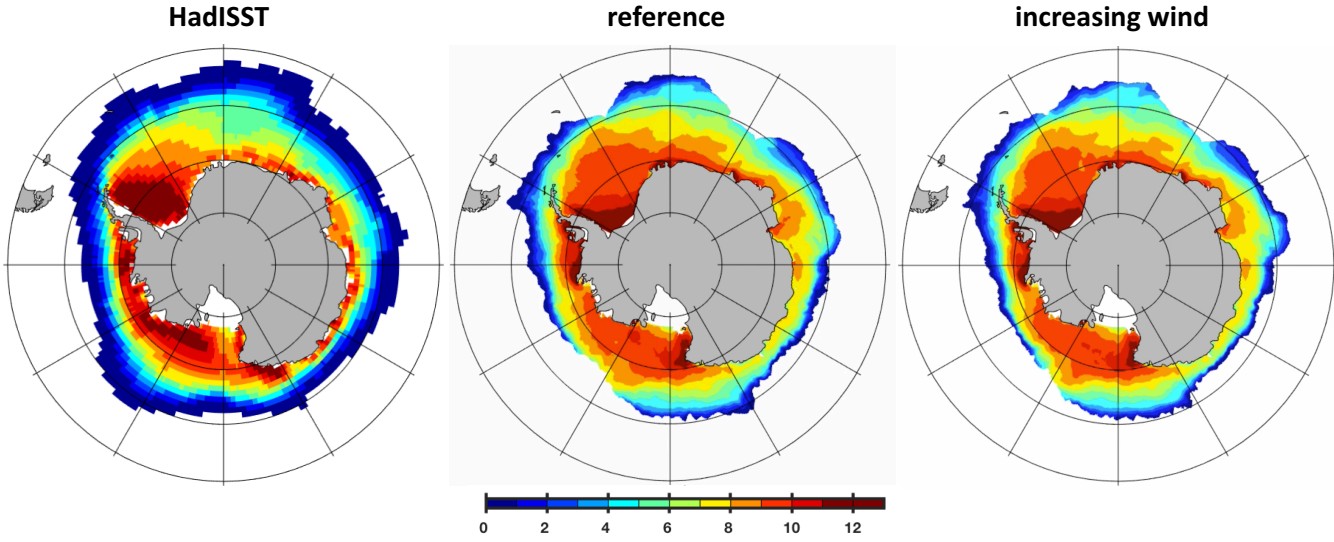

**Figure 23.** Ice-covered months in a year. The left panel refers to a 1980 to 2000 average based on the HadISST observational estimate (Rayner et al., 2003). The middle and right panel refer to the nominal year 2022 of the reference and increasing-wind simulation, respectively.

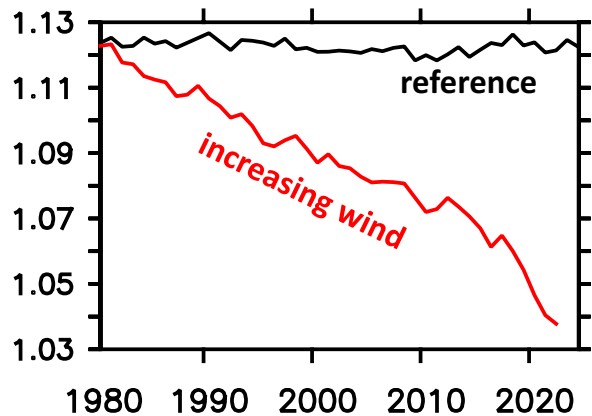

**Figure 24.** Simulated annual mean sea ice cover south of $40°$S in units $10^7\,km^2$. The black (red) line refers to the reference (increasing-wind) simulation.

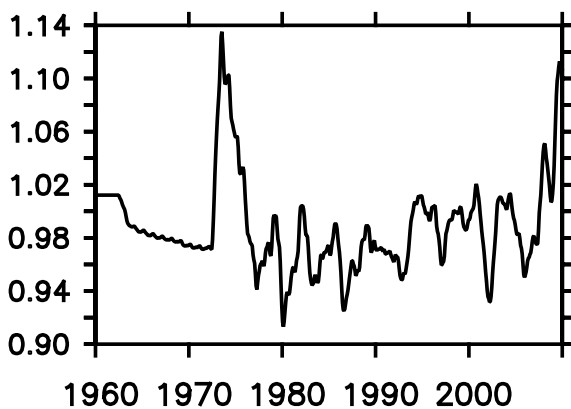

**Figure 25.** Observational estimate of annual mean sea ice cover south of $40°$S in units $10^7\,km^2$ calculated from HadISST (Rayner et al., 2003).

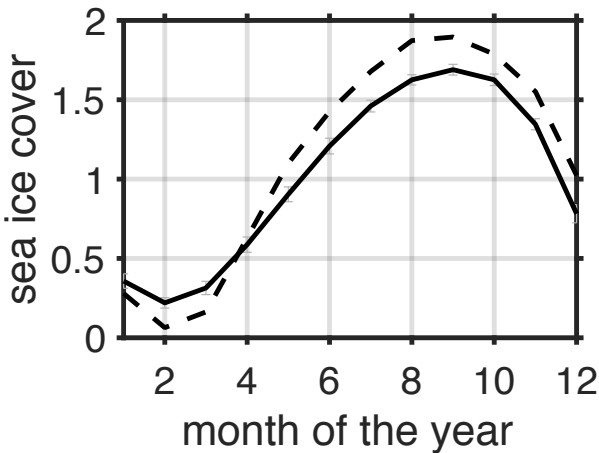

**Figure 26.** Simulated (dashed line, reference simulation) and observed (solid line, reanalysis from NSIDC) climatological annual cycle of sea ice cover south of $40°$S in units $10^7\,km^2$.

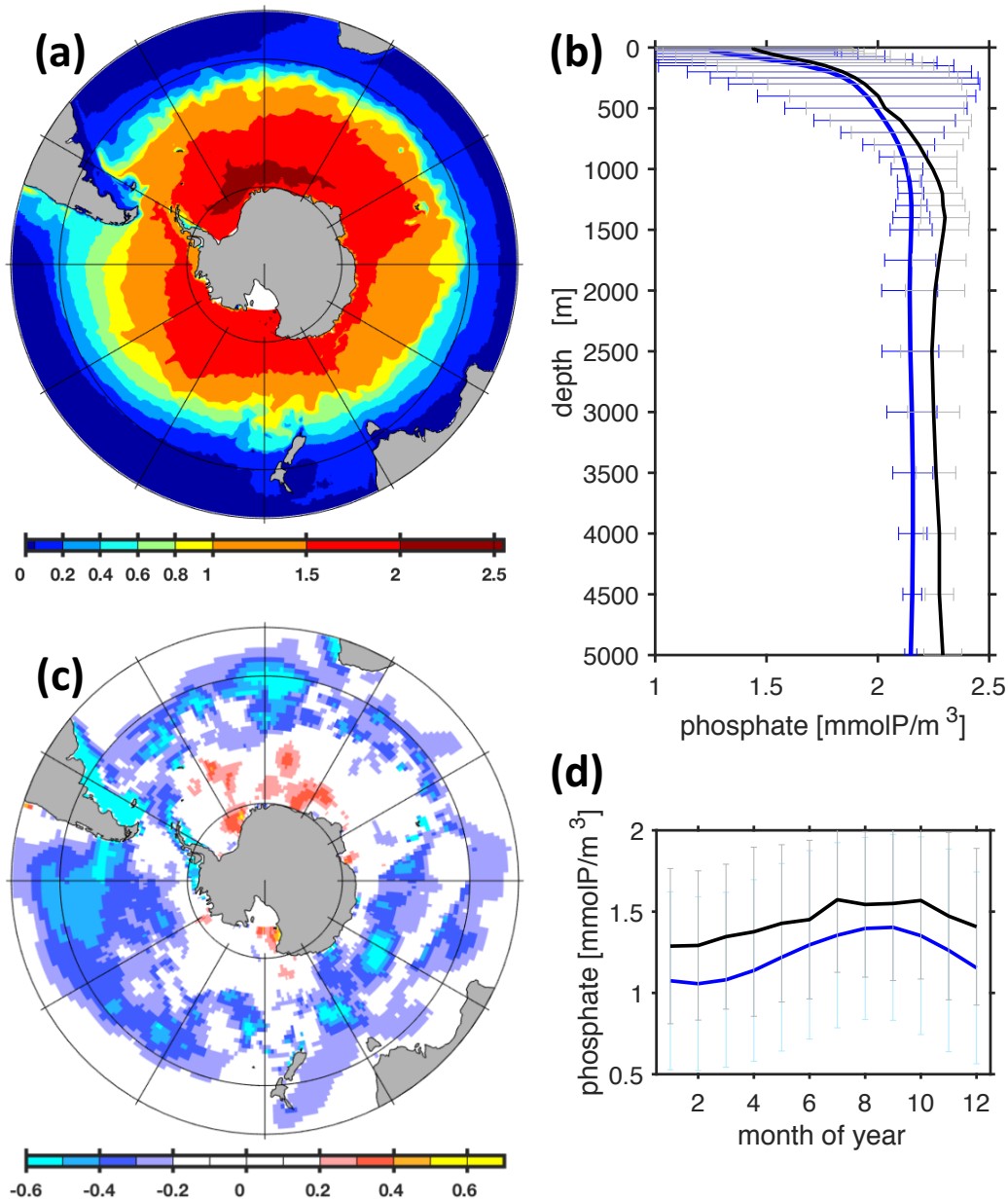

**Figure 27.** Comparison of simulated and observed phosphate concentrations. Panel **(a)** refers to simulated surface phosphate concentrations (reference simulation, averaged over nominal years 1993 to 1998). The colorbar denotes phosphate concentrations in units $mmol\,P\,m^{-3}$. Panel **(b)** shows simulated (blue line) and observed (black line) phosphate concentrations averaged horizontally in the Southern Ocean (i.e. south of $40°$S) along with their respective spatial standard deviations (grey and blue horizontal bars). Pannel **(c)** shows the difference between simulated and observed (Garcia et al., 2010) surface concentrations. Panel **(d)** shows the simulated (blue line) and observed (black line) seasonal cycle of dissolved phosphate concentration at the surface calculated as monthly means averaged over the Southern Ocean.

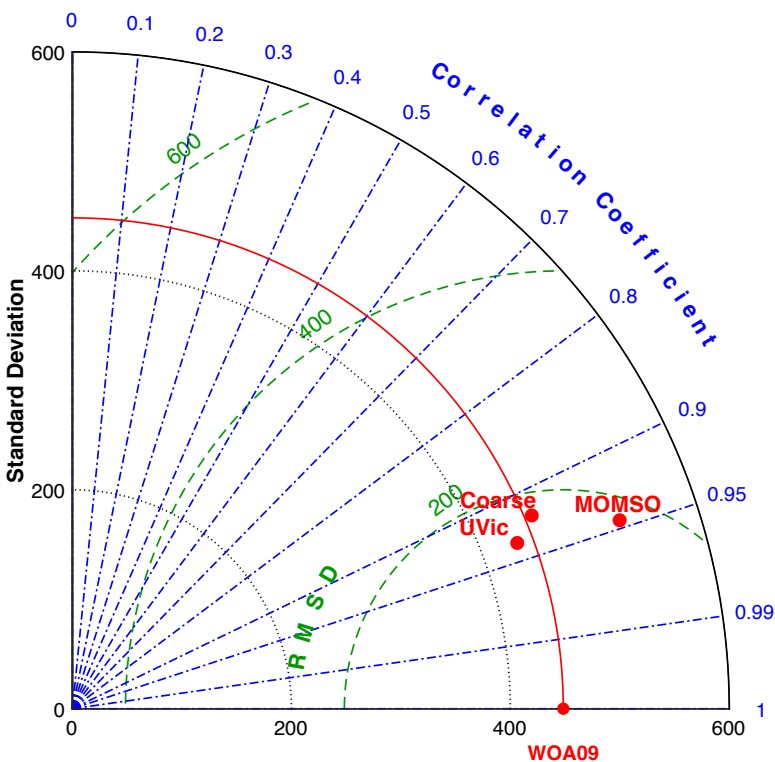

**Figure 28.** Taylor diagram of simulated phosphate concentration referenced to Southern Ocean WOA09 (Garcia et al., 2010) annual mean climatology. The units for the standard deviation and root-mean-square deviations are $\mu\,mol\,P\,m^{-3}$. *Coarse* refers to the very similar albeit coarser, non-eddying reference configuration used in Dietze et al. (2017) and *UVic* refers to the non-eddying Earth System Model used e.g. in climate engineering assessments (c.f. reference version in Löptien and Dietze, 2019; Keller et al., 2012)

.

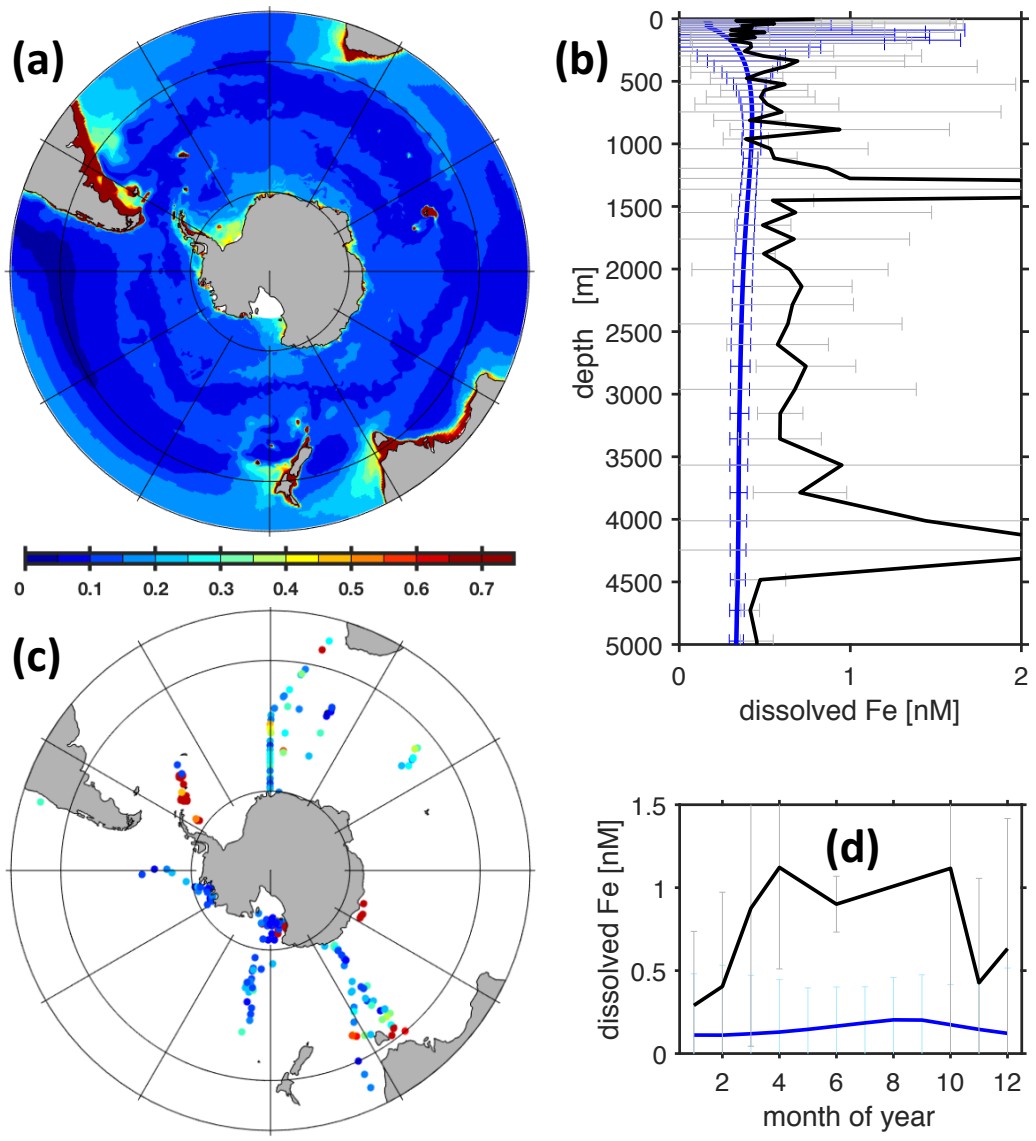

**Figure 29.** Comparison of simulated with observed dissolved iron concentrations. Panel **(a)** refers to simulated surface iron concentrations (reference simulation, averaged over nominal years 1993 to 1998). The colorbar denotes iron concentrations in units $nM\ Fe$. Panel **(b)** shows simulated (blue line) and observed (black line) concentrations averaged horizontally in the Southern Ocean (i.e. south of $40°$S) along with their respective spatial standard deviations (grey and blue horizontal bars). Panel **(c)** refers to observed surface iron concentrations (compiled by Tagliabue et al. (2012) & Mawji, E., et al. (2014)). (The colorbar matches Panel a). Panel **(d)** depicts the simulated (blue line) and observed (black line) seasonal cycle of dissolved iron concentration at the surface calculated as monthly means averaged over the Southern Ocean.

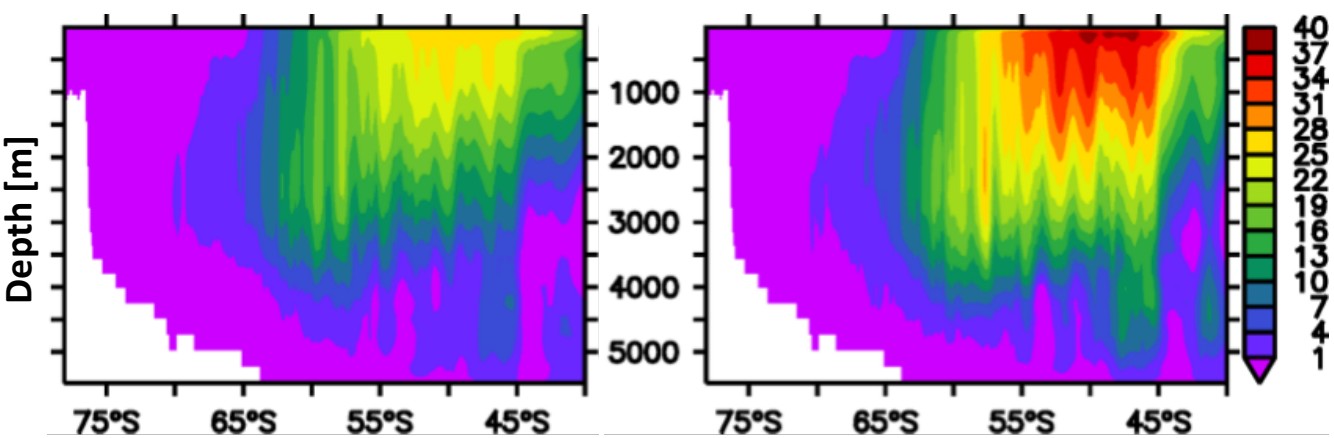

**Figure 30.** Simulated Southern Ocean meridional overturning circulation in units $10^6\,m^3\,s^{-1}$ averaged over nominal year 2024. The left (right) panel refers to the reference (increasing WIND) simulation.

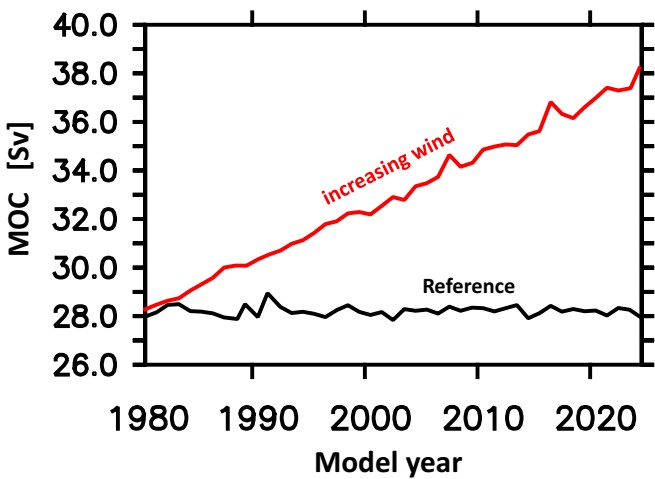

**Figure 31.** Maximum Southern Ocean Eularian meridional overturning circulation south of $40°$S in units $10^6\,m^3\,s^{-1}$. The black (red) line refers to the reference (increasing-wind) simulation.

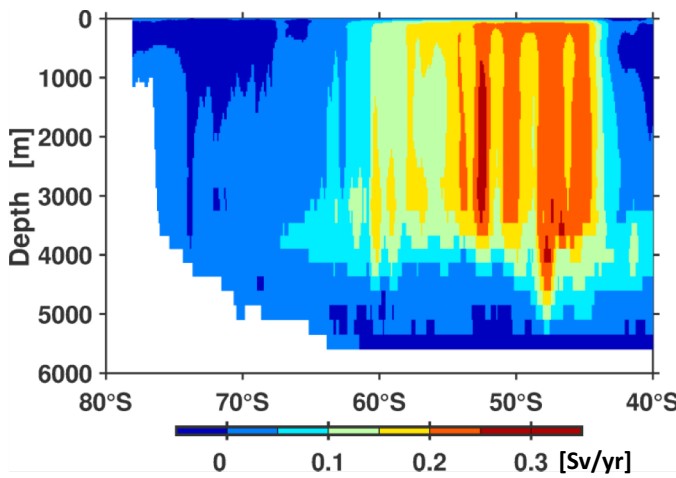

**Figure 32.** Simulated linear trend of Southern Ocean Eularian meridional overturning circulation effected by increasing winds during nominal years 1980 to 2024 in units $10^6 \, m^3 \, s^{-1} \, yr^{-1}$. Positive values denote increasing overturning.

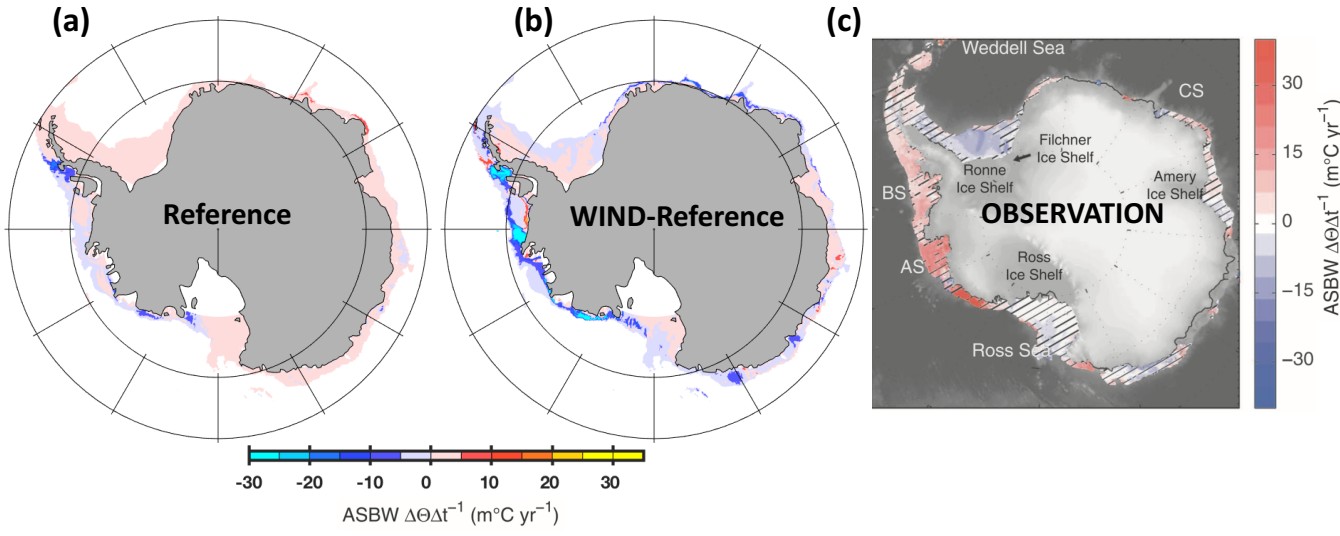

**Figure 33.** Temporal trend in the temperature of the Antarctic Continental Shelf Bottom Water (ASBW) at the seabed for depths shallower than 1500 m. Panel **(a)** refers to the drift that is still persistent in the reference simulation during nominal years 1980 to 2024. **(b)** refers to the trend in simulation in the simulation with increasing winds corrected by the drift that still persists in the reference. Panel **(c)** refers to an observational estimate compiled by Schmidtko et al. (2014).

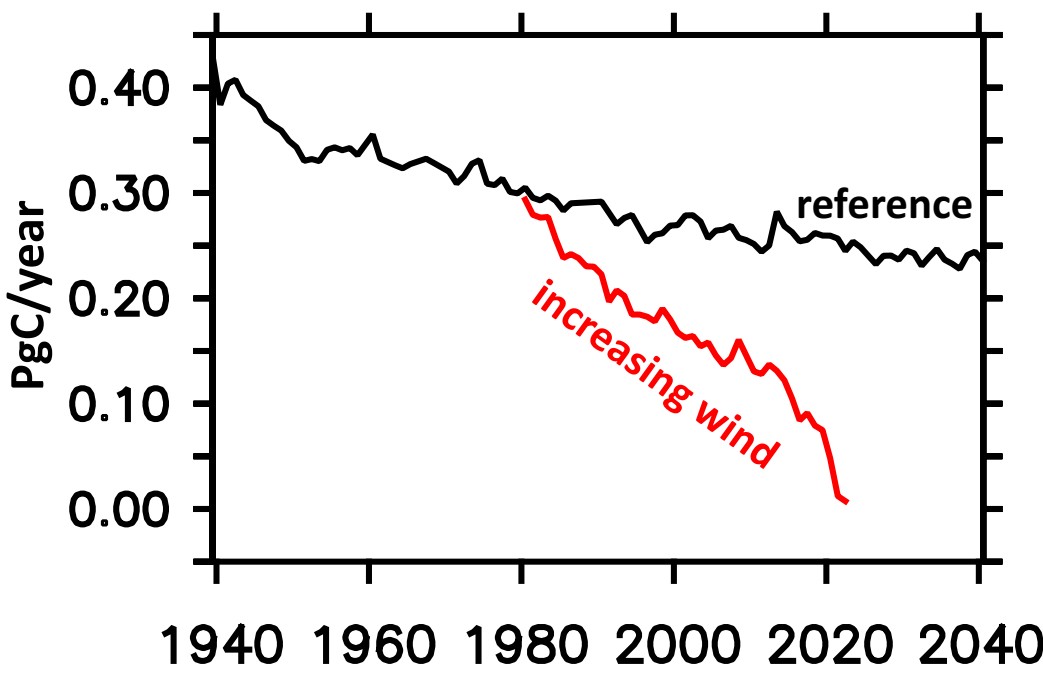

**Figure 34.** Simulated oceanic uptake of natural carbon south of $40°$S. The black (red) line refers to the reference (increasing-wind) simulation.

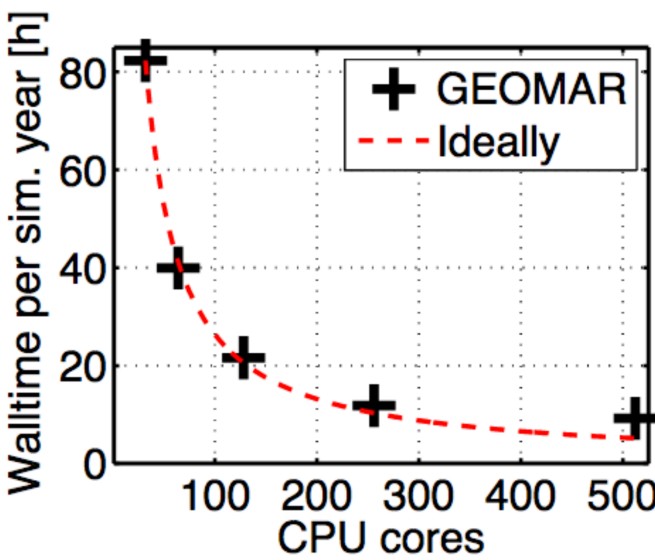

**Figure 35.** Computational performance as a function of CPU cores put to work simultaneously. On 500 Intel Xeon CPU E5-2670 cores it takes 10 hours to simulate one year.