# Peer review of "MOMSO 1.0 - an eddying Southern Ocean model configuration with fairly equilibrated natural carbon"

_Geoscientific Model Development, 2018_

## Referee Comment (RC1) · Anonymous Referee #1 · 6 Feb 2019

This paper documents the development of an interesting model configuration designed to allow for an eddying simulation of the Southern Ocean that also includes biogeo-chemistry. The model represents a series of compromises that mean whilst none of the components are ideal, the whole produces a potentially useful model simulation. The meat of the paper is validation of the physical circulation. This section requires some work, as there is considerable detail missing and a lack of quantification. The closing section of the paper looks at the biogeochemistry in the model's reference run and a short experiment with increased winds. This shows the capability of the model

produce useful biogeochemical results, despite the physical circulation not being fully equilibrated. However, there is also a lack of detail here, which presents the full power of the modelling approach from being illustrated.

Major Comments

page 2, lines 4-9 : This is not an accurate representation of current thinking on the overturning circulation of the Southern Ocean. There is no debate as to whether the bolus overturning due to eddy fluxes of thickness opposes the so-called Deacon cell. It is extremely well established that it does so, to the degree that the Southern Ocean literature does not discuss the Deacon cell and instead talks about Eulerian overturning and bolus overturning. There is an on-going debate regarding the degree to which the bolus overturning cancels out the Eulerian, and that to which their residual responds to forcing changes, particularly with respect to wind stress changes, and model differences. However, that this cancellations takes place is widely accepted. See, e.g., Marshall & Radko (2003), Viebahn & Eden (2010), and Abernathey et al. (2011), etc.

page 2, line 24 onwards : There is at least one example of a fully-equilibrated, both thermodynamically and biogeochemically, model study that the authors could cite here; Munday et al. (2014). Whilst the model in Munday et al. is too coarse to be truly eddy resolving, it does have substantial internal variability and large-scale vortices, which leads to changes in sensitivity to wind stress of the physical circulation consistent with higher resolution models.

Section 2.1 : Also of importance is that, since z* is essentially an extension of the nonlinear free surface method of Campin et al. (2004) to all model levels, it also gives very accurate conservation of tracers.

page 5, lines 4-5 : The Smagorinsky scheme has a single coefficient, usually taken to be in the range 2-4, that allows some control over the viscosity. Please add the coefficient value you chose here. In addition, the issues arising near steep topography suggests that this coefficient was too small. Is there a reason why the authors chose to

introduce additional viscosity instead of just increasing the viscosity value? Additionally, some of the issues could result from the use of purely Laplacian viscosity. Given the grid scale variance at the resolution of MOMSO, it is probably appropriate to start applying some biharmonic viscosity/diffusion to the model.

page 5, lines 5-7 : The PPM scheme is a good choice for the advection of temperature/salinity. The choice of advection scheme for the biogeochemistry is also very important (see Levy et al. 2001). Having different schemes for these two choices could have repercussions. Have the authors investigated this?

Section 2.5 : Initialising from a previous stratification/tracer distribution can be a very good way to shorten the spinup of high resolution models. Is there a reason why the temperature and salinity from the same run that produced the biogeochemical tracers wasn't also used?

Section 3.1 : A general criticism of this section is that it is largely qualitative, even when quantitative comparisons could be carried out, e.g. the spatial pattern and values of EKE/SSH variance are available from the altimetry the authors use. A properly quantitative comparison would allow more detail to be drawn out. In addition, the authors highlight the surface velocities and EKE as having "remarkable" comparison to the altimetry. This is hardly remarkable; the surface velocities and EKE look largely as one would expect for a model of $\sim$1/5o grid spacing in the Southern Ocean (see, e.g. Delworth et al., 2012, or Barnier et al., 2006, for examples of how similar resolution ocean models, or ocean components of coupled models, look).

In addition, the deficiencies in particular comparisons are thoroughly glossed over. Something that stood out to me is that the flow is too zonal and there is too much EKE south of Australia. The East Australia Current is also poorly represented and the Agulhas retroflection has too small a region of high EKE. I'd expect the mean flow and eddies south of Australia to be related to the high temperatures south of this region shown in Figure 9.

Most of the verification of the model is carried out on the surface quantities. Whilst this is appropriate given the length of the model run, which means that below the mixed layer there might not have been a great deal of adjustment, it is still useful to consider it briefly as it would place the later biogeochemical validation in context. At the very least, they should look at some temperature/salinity/density/biogeochemical transects across the Southern Ocean and the pycnocline depth. This will likely tie into the noted low value of the ACC transport.

There is no discussion of the mixed layer depth. This is fairly well observed in recent years thanks to Argo floats and would be a key parameter for the exchange of carbon with the atmosphere and for the subduction of biogeochemical tracers from the surface.

Section 3.2 : Unfortunately, this section is completely deficient. There are many ways to calculate overturning (see discussion above about Eulerian vs. residual vs. bolus overturning) and there is no information given here as to what variety of overturning is presented in Figure 23. It should be some measure of the residual circulation, although given the increase in the WIND simulation it looks like it might be the Eulerian overturning. A better comparison would be to calculate the residual overturning in density coordinates and compare the residual overturning, as well as the Eulerian and bolus overturning. This would give a much better sense of how the overturning is changing between the two experiments and how much the remaining drift in REF is affecting their results. If the subsurface stratification is changing, then it will make interpretation more difficult, since the there could be significant diapycnal transformations in the streamfunction. This is another reason to look at the subsurface stratification earlier in the paper.

Minor Comments

page 1, line 20 : The choice of 40oS is not well justified, why not 35oS or 45oS? There are physical aspects of the circulation that could be cut off by this choice, for example the region in which the Agulhas current interacts with the ACC is very close to 40oS.

page 6, line 5-8 : Do the climatological estimates include the seasonal cycle?

Section 3.1.4 : A missing quantitative comparison here is between the flux due to the surface restoring of salinity and the other surface fluxes that contribute to salinity changes.

Figures such as 14 & 15, where the reader is invited to compare the REF and WIND experiments variation with the observed quantity would work much better as a single figure. Preferably as a single panel, with the observations as a third line.

Section 3.1.6 : Based on Figures 19 & 20, the average concentration of both iron and phosphate appears low. Is this because the initial conditions for these fields are biased low? And if so, could the issues raised in this section be improved by simply adding more phosphate & iron?

page 12, line 26 : In what sense to the compiler optimisations break reproducibility? Will the same model year, run on the same machine, be different if the model was rerun? Or will it only change when run on a different machine with different compilers? The first is rather alarming, the second would be very common.

page 13, line 12 : Quantum leap is an overstatement, given the length of the physical model spinup, the still low resolution of the physical model and the documented deficiencies of both physical and biogeochemical system.

Technical Issues

page 2, lines 13-14 & 27 : Work in progress for whom? The authors or the community in general, it could be either!

page 2, line 14 : Put comma after progress and delete comma from after extent.

page 2, line 18 : "As for now we know"?

page 3, line 11 "by chance", isn't it more by design?

page 3, line 13 : "allows to test" -> "allows us to test".

page 4, line 9, page 7, line 10 : Orphan lines like this pop up quite a bit throughout the paper.

page 4, line 19 : 42.429.759 -> 42 429 759

page 5, line 15 : I think the authors mean vicious cycle! (Even though viscous seems appropriate for numerical models)

page 5, line 24 : Galbreith -> Galbraith

page 8, line 5 : Spacial is an accepted alternative to the more common spatial spelling. However, the paper uses both spellings at different places.

page 10, lines 21-24 : Is the reference to "module" meaning a specific part of the numerical code? or an inaccuracy in the overall representation of part of the system?

page 12, line 16 : There's something wrong with the brackets here.

Figure 23 : noetig? in the caption.

References

Abernathey, R., J. Marshall, and D. Ferreira, 2011: The dependence of Southern Ocean meridional overturning on wind stress. J. Phys. Oceanogr., 41, 2261–2278.

Barnier, B., G. Madec, T. Penduff, J. M. Molines, A. M. Treguier, J. Le Sommer, A. Beckmann, A. Biastoch, C. Boning, J. Dengg, C. Derval, E. Durand, S. Gulev, E. Remy, C. Talandier, S. Theetten, M. Maltrud, J. McClean, and B. de Cuevas, 2006: Impact of partial steps and momentum advection schemes in a global ocean circulation model at eddy-permitting resolution. Ocean Dyn., 56, 543–567.

Campin, J. M., A. Adcroft, C. Hill, and J. Marshall, 2004: Conservation properties in a free-surface model. Ocean Modell., 6, 221–244.

Delworth, T. L., A. Rosati, W. Anderson, A. J. Adcroft, V. Balaji, R. Benson, K. Dixon,

S. M. Griffies, H.-C. Lee, R. C. Pacanowski, G. A. Vecchi, A. T. Wittenberg, F. Zeng, and R. Zhang, 2012: Simulated climate and climate change in the GFDL CM2.5 high-resolution coupled climate model. J. Clim., 25, 2755–2781.

Levy, M., A. Estublier, and G. Madec, 2001: Choice of advection scheme for biogeo-chemical models. Geophys. Res. Lett., 28, 3725–3728.

Marshall, J. and T. Radko, 2003: Residual-mean solutions for the Antarctic Circumpolar Current and its associated overturning circulation. J. Phys. Oceanogr., 33, 2341–2354.

Munday, D. R., H. L. Johnson, and D. P. Marshall, 2014: Impacts and effects of mesoscale ocean eddies on ocean carbon storage and atmospheric pCO2. Global Biogeochem. Cycles, 28, 877–896, doi:10.1002/2014GB004836.

Viebahn, J. and C. Eden, 2010: Towards the impact of eddies on the response of the Southern Ocean to climate change. Ocean Modell., 34, 150–165.

---

## Referee Comment (RC2) · Anonymous Referee #2 · 29 May 2019

General comments: This paper presents a new ocean model configuration, which couples an ecosystem model to an active ocean-sea ice model, with a refined horizontal resolution in the Southern Ocean. The stated aim of this new configuration is to examine the Southern Ocean carbon budget and its sensitivity to wind changes. Although this new model configuration is promising, this paper fails to provide quantitative evidence of the added value of this configuration, and lacks adequate discussion of the applicability and limitations of the technical choices made or the resulting biases.

Most results included are side-by-side plots of surface metrics in the simulation and

observations. The model evaluation needs to include quantitative metrics and/or plots of differences, and would benefit from comparisons with existing reanalysis products or state estimates. Biases are only briefly (if at all) mentioned and there is generally no attempt to identify (and provide evidence for) the possible causes of these biases. In addition, the manuscript mentions a 'twin' configuration with coarser resolution, but does not include any comparison with that existing configuration (which would significantly improve the manuscript). The objectives of the wind sensitivity experiment are not clearly stated in the introduction, and the limitations of this experiment need to be discussed any time its results are included. This manuscript also needs to include a more detailed discussion about the applicability of the results and the limits of this configuration.

Specific comments:

Abstract: The abstract makes the case that the model 1) combines a carbon inventory and eddy-permitting resolution in a way that outperforms existing models, 2) is suited to sensitivity experiments at decadal timescales, 3) can provide boundary condition to ice-sheet models. This manuscript would benefit from focusing the analysis and figures to demonstrate these 3 points, and from including a discussion section on the limitations of the model with regards to those 3 points.

Introduction: Overall, this section would benefit from a more focused approach, targeting specific dynamics or metrics that will be covered in the body of the paper.

Page 1, Line 20: There is no justification as to why 40S was chosen, or even simply how it relates to the above discussion about the location of the STF. Page 2, 29-31: Please consider how this fits within other work of Southern Ocean response to winds, such as (Spence et al, 2010). Page 3, 14: Where is this comparison between coarse and finer resolution model included? The manuscript would benefit from one. Page 3, 15-22: the discussion of the role of meltwater on ocean circulation or biogeochemistry is out of place here. Given that the current configuration does not include sub-ice-shelf

cavities or parameterized meltwater fluxes, it will not be able to account for meltwater feedbacks. At best, this could be included in a discussion section to discuss options for further development. Page 4, line 15: Some justification on why topography needs to be filtered 3 times (even just a qualitative target) is needed here. Page 5, line 8-11: I have some concerns about the increase of viscosity over the Drake Passage. Some justification and possible root causes of this problem over Drake Passage need to be included.

Section 2.3 This section is too short and vague. If a change was made to the standard code and led to different results, please show the impact that this change has made (in an Appendix if necessary).

Section 2.4 Design: please specify "complex-enough" for what? Or the model fidelity in representing what? At least include the target variables and their scientific relevance. Why is the comparison with the coarse-resolution model not included in this paper? This manuscript needs to be self-sufficient.

Section 2.6 Please include the frequency (or time scale) of boundary restoring. Also, the boundary restoring is necessary because the domain does not include dense water formation regions (it is not simply lack of processes). It is unclear what the wind experiment is supposed to address. If it is supposed to mirror the historical trend, please address why the physics are set to a historical trend while the carbon is maintained at a pre-industrial level. The objective of this experiment needs to be set up here, to provide context for the interpretation of results and discussion of their applicability.

Section 3. Why does this section not include any comparison with reanalysis data or state estimates? These datasets are commonly used in the literature, and provide more comprehensive data (sub-surface, time-varying...), which would help make a more compelling case for the fidelity of this new model configuration. As it is, this section appears extremely light and unconvincing.

Section 3.1.1. This section lack qualitative evidence of the model's ability to represent the ocean circulation metrics mentioned. It is necessary to define the meridional overturning as calculated here. Without a clear definition, the quantitative results are difficult to interpret or compare to other studies.

Section 3.1.2 This section lack qualitative evidence of the model's ability to produce 'realistic' levels of EKE. Given that this is a main point (included in the paper's title), putting plots side-by-side (Fig 8) is far from enough. Please include a plot showing the difference between the two datasets, and a quantification of mean EKE per region to provide some level of quantitative assessment. In addition, some discussion of the results is needed: what are some of the biases present? What may cause these biases? Finally, the conclusion that a 'realistic' level of EKE necessarily equates a good representation of eddy-driven processes is simplistic. Here, the only metric is surface EKE, whereas eddy-driven processes, (including eddy-driven upwelling of nutrient-rich waters) occur over a range of depths. There is a vast body of literature investigating eddy processes in the Southern Ocean and what resolution may be necessary/sufficient to represent them adequately. Some discussion of this literature, and of how this particular model configuration fits in the context of other modeling studies is necessary.

Section 3.1.3 Again, some quantitative comparison between the observation-based datasets and the modeled values is needed (e.g. plot of the difference), as well as a discussion of the biases (especially between the bottom temperature values, which has no discussion of biases at all). Combining Fig 10 and 11 would help the comparison.

Section 3.1.5 A more comprehensive assessment of sea ice would make a more compelling case (e.g. sea ice concentration, sea ice thickness, annual cycle of sea ice area).

Section 3.1.6 This section needs a more comprehensive assessment of the model's performance in representing observed patterns of biogeochemical properties. A comparison to the Biogeochemical Southern Ocean State Estimate (B-SOSE) (Verdy and Mazloff, 2017) would be a good step forward. In addition, the biogeochemical performance of this model configuration should be shown (not just said) to be comparable to the one from the existing coarse resolution model. Likewise, there is little discussion of the possible causes of the biases, and lack of evidence to support the possible causes mentioned.

Section 3.2 It is unclear what this sensitivity experiment is for, and why only the winds were changed. It lacks discussion of the mechanisms leading to the change in overturning circulation, or to the change of bottom water temperature. For the carbon results, it should be specified that the change in the experiment is showing only the impact of the physical adjustment to winds (given that the carbon concentration is maintained to pre-industrial levels).

Section 4: summary and conclusions This section makes qualitative statements about the model's fidelity, which have not been adequately supported by the body of the paper (similar to the abstract). As the model performance with respect to biogeochemistry is said to be similar to the coarser configuration, it is essential to demonstrate the benefits of this configuration with respect to eddy processes. Describing the configuration as a 'quantum leap' in modeling sounds over-reaching, given the lack of quantitative evidence included in the current manuscript to demonstrate the improved performance of this model configuration compared to existing configurations. This section needs to include an in-depth discussion of the relevance and applicability of this model configuration. For example, the time scales examined here are the decadal timescales, while modeling of ice-shelf melt or Antarctic Bottom Water are more relevant to longer time scales.

Technical Corrections:

Page 2, line 10: "effects and affects" Line 12 budged → budget Line13 net-air → net air Line 25 'impinges' → impacts/affects Page 3, line 19: fuel → lead to Page 4, line 6: biogeochemical Page 5, line 23: it's fidelity → its

References: A. Verdy and M. Mazloff, 2017: " A data assimilating model for estimating Southern Ocean biogeochemistry." J. Geophys. Res. Oceans., 122, doi:10.1002/2016JC012650.

Spence, P. , J. C. Fyfe , A. Montenegro , and A. J. Weaver , 2010: Southern Ocean response to strengthening winds in an eddy-permitting global climate model. J. Climate, 23, 5332– 5343, https://doi.org/10.1175/2010JCLI3098.1.

———————————

---

## Author Comment (AC1) · 12 Jul 2019

A: Referee #1 asks for more details (including quantitative measures) in the validation of the physical circulation. Further he is interested in details regarding the biogeochemistry in the model's reference run and short sensitivity runs. In addition, he wants to see a more comprehensive discussion of the "overturning concept" in the introduction.

The specific comments of Referee #1 are:

**Major Comments**
**R: page 2, lines 4-9 : This is not an accurate representation of current thinking on the overturning circulation of the Southern Ocean. There is no debate as to whether the bolus overturning due to eddy fluxes of thickness opposes the so-called Deacon cell. Itis extremely well established that it does so, to the degree that the Southern Ocean literature does not discuss the Deacon cell and instead talks about Eulerian overturning and bolus overturning. There is an on-going debate regarding the degree to which the bolus overturning cancels out the Eulerian, and that to which their residual responds to forcing changes, particularly with respect to wind stress changes, and model differences. However, that this cancellations takes place is widely accepted. See, e.g.,Marshall & Radko (2003), Viebahn & Eden (2010), and Abernathey et al. (2011), etc.**

A: We will include a more comprehensive discussion of the overturning in the introduction of our model description paper and cite the papers proposed by the reviewer in the revised version of the manuscript.

**R: page 2, line 24 onwards : There is at least one example of a fully-equilibrated, both thermodynamically and biogeochemically, model study that the authors could cite here; Munday et al. (2014). Whilst the model in Munday et al. is too coarse to be truly eddy resolving, it does have substantial internal variability and large-scale vortices, which leads to changes in sensitivity to wind stress of the physical circulation consistent with higher resolution models**

A: We will add the respective reference to in the revised version of the manuscript.

**R: Section 2.1 : Also of importance is that, since z\* is essentially an extension of the nonlinear free surface method of Campin et al. (2004) to all model levels, it also gives very accurate conservation of tracers.**

A: Thanks for pointing this out. We will highlight this advantage of our choice of numerics accordingly in the revised version of the manuscript.

**R: page 5, lines 4-5 : The Smagorinsky scheme has a single coefficient, usually taken to be in the range 2-4, that allows some control over the viscosity. Please add the coefficient value you chose here. In addition, the issues arising near steep topography suggests that this coefficient was too small. Is there a reason**

**why the authors chose to introduce additional viscosity instead of just increasing the viscosity value? Additionally, some of the issues could result from the use of purely Laplacian viscosity. Given the grid scale variance at the resolution of MOMSO, it is probably appropriate to start applying some biharmonic viscosity/diffusion to the model.**

A: We will add the information on the coefficient in the revised version of the manuscript. Further, we will elaborate on our attempts to fix the problem which indeed included biharmonic diffusion and/or viscosity, overall higher Smagorinski coefficient, decreased winds, changed bottom topography and changes to the vertical diffusion/friction. We really went at it - but did not list all of our failed attempts to fix the problem in the original version of the manuscript.

**R: page 5, lines 5-7 : The PPM scheme is a good choice for the advection of temperature/salinity. The choice of advection scheme for the biogeochemistry is also veryimportant (see Levy et al. 2001). Having different schemes for these two choices could have repercussions. Have the authors investigated this?**

A: No, we have not investigated this. But we agree that advection numerics adds uncertainty. We will state that in the revised version of the manuscript. We also agree that a comprehensive study on the effects of advection numerics would be very interesting. We expect repercussions even if the same scheme is used for all (passive and active) tracers because spurious numerical effects are dependent on tracer gradients and those differ between temperature and salinity which affect the saturation state of carbon dioxide and DIC on the other hand. We will add a paragraph on this issue in the revised version of the manuscript.

**R: Section 2.5 : Initialising from a previous stratification/tracer distribution can be a very good way to shorten the spinup of high resolution models. Is there a reason why the temperature and salinity from the same run that produced the biogeochemical tracers wasn't also used?**

A: 3-D model initialisation is indeed an issue and we will add a paragraph on this subject in the revised version of the manuscript. Ideally, we would have liked to start all prognostic variables of the high-resolution model from observations (and we would have liked the high-resolution model to stay close to these initial conditions). Unfortunately, observational data coverage is very sparse for the biogeoschemical tracers (especially for iron). Thus, we decided to opt for a model-based product. Since we planned a comparison with a coarse resolution model anyways we decided to use the coarse resolution model output as input. In a way, we explore how a coarse-resolution biogeochemical model state "performs" in a model with "enhanced" (higher resolution) physics.

**R: Section 3.1 : A general criticism of this section is that it is largely qualitative, even when quantitative comparisons could be carried out, e.g. the spatial pattern and values of EKE/SSH variance are available from the altimetry the authors use. A properly quantitative comparison would allow more detail to be drawn out. In addition, the authors highlight the surface velocities and EKE as having "remarkable" comparison to the altimetry. This is hardly remarkable; the surface velocities and EKE look largely as one would expect for a model of~1/5o grid spacing in the Southern Ocean (see, e.g. Delworth et al., 2012, or Barnier et al., 2006, for examples of how similar resolution ocean models, or ocean components of coupled models, look). In addition, the deficiencies in particular comparisons are thoroughly glossed over. Something that stood out to me is that the flow is too zonal and there is too much EKE south of Australia. The East Australia Current is also poorly represented and the Agulhas retroflection has too small a region of high EKE. I'd expect the mean flow and eddies south of Australia to be related to the high temperatures south of this region shown in Figure 9.**

A: (1) concerning "remarkable EKE comparison to altimetry": This is a misunderstanding that we will fix in the revised version of the manuscript. It is true, the configuration presented here is featuring a performance comparable to other high-resolution configurations. The selling point here is that the configuration has a sufficiently-spun-up biogeochemical cycle to address some carbon-related questions.

I started my career with 1/3 degree models and I am still amazed by the remarkable realism of (almost) ALL high-resolution models.

(2) concerning "too much EKE south of Australia/East Australia Current/ Aghulas retroflection": We will include a more detailed discussion in the revised version of the manuscript. E.g. the East Australia Current is already affected by the coarser resolution which presides outside our zone of interest (i.e. outside the Southern Ocean).

**R:  Most of the verification of the model is carried out on the surface quantities. Whilst this is appropriate given the length of the model run, which means that below the mixed layer there might not have been a great deal of adjustment, it is still useful to consider it briefly as it would place the later biogeochemical validation in context. At the very least, they should look at some temperature/salinity/density/biogeochemical transects across the Southern Ocean and the pycnocline depth. This will likely tie into the noted low value of the ACC transport. There is no discussion of the mixed layer depth. This is fairly well observed in recent years thanks to Argo floats and would be a key parameter for the exchange of carbon with the atmosphere and for the subduction of biogeochemical tracers from the surface.**

A: We will add a meridional section of T/S and PO4. Further we will show a comparison with surface mixed layer depth.

**R: Section 3.2 : Unfortunately, this section is completely deficient. There are many ways to calculate overturning (see discussion above about Eulerian vs. residual vs. bolusoverturning) and there is no information given here as to what variety of overturning is presented in Figure 23. It should be some measure of the residual circulation, although given the increase in the WIND simulation it looks like it might be the Eulerian over-turning. A better comparison would be to calculate the residual overturning in density coordinates and compare the residual overturning, as well as the Eulerian and bolus overturning. This would give a much better sense of how the overturning is changing between the two experiments and how much the remaining drift in REF is affecting their results. If the subsurface stratification is changing, then it will make interpretation more difficult, since the there could be significant diapycnal transformations in the streamfunction. This is another reason to look at the subsurface stratification earlier in the paper.**

A: We will clarify the computation of overturning in the revised version of this manuscript. A comprehensive study between the experiments WIND and REF is in preparation but beyond the scope of this manuscript. This GMD manuscript is a first step. It is dedicated to describing the model setting and hinting at potential projects which could be carried out within such a model framework. We will clarify this in the revised version of the manuscript.

**Minor Comments**

**R: page 1, line 20 : The choice of 40oS is not well justified, why not 35oS or 45oS? Thereare physical aspects of the circulation that could be cut off by this choice, for example the region in which the Agulhas current interacts with the ACC is very close to 40oS.**

A: We will clarify that the choice of 40S has been motivated by other studies (to some of which we compare our results) using this exact threshold. We agree that this choice is arbitrary and not suitable for all purposes.

**R: page 6, line 5-8 : Do the climatological estimates include the seasonal cycle?**

A: We will add the missing information in the revised version of the manuscript.

**R: Section 3.1.4: A missing quantitative comparison here is between the flux due to the surface restoring of salinity and the other surface fluxes that contribute to salinity changes.**

A: Agreed, we will add the respective information in the revised version of the manuscript.

**A: Figures such as 14 & 15, where the reader is invited to compare the REF and WIND experiments variation with the observed quantity would work much better as a single figure. Preferably as a single panel, with the observations as a third line.**

A: Agreed we will change that in the revised version of the manuscript.

**R: Section 3.1.6 : Based on Figures 19 & 20, the average concentration of both iron and phosphate appears low. Is this because the initial conditions for these fields are biased low? And if so, could the issues raised in this section be improved by simply adding more phosphate & iron?**

A: I guess so, and yes, I guess this could be improved by changing the initial conditions - however - at a price:  an increased drift towards just these low biased conditions. We will look into this and add a discussion in the revised version of the manuscript.

**R: page 12, line 26 : In what sense to the compiler optimisations break reproducibility? Will the same model year, run on the same machine, be different if the model was rerun? Or will it only change when run on a different machine with different compilers?The first is rather alarming, the second would be very common.**

A: Yes, the former (depending on compile options). This is well known for some (very performant) compiler settings and we have tested the effect of this in a similar setup. We will include more information on this issue (including some links to intel's compiler documentation) in the revised version of the manuscript.

**R: page 13, line 12 : Quantum leap is an overstatement, given the length of the physical model spinup, the still low resolution of the physical model and the documenteddeficiencies of both physical and biogeochemical system.**

A: Sorry, for the offensive phrasing - we will change it in the revised version of the manuscript. All "free" (i.e. not data assimilating) eddy-resolving ocean-circulation biogeochemical Southern Ocean model configurations we are aware of are of limited use in terms of simulating decadal changes of Southern Ocean carbon inventories because of either (1) they are regional configurations that are affected by prescribed spatial boundary conditions or (2) they could not be integrated long enough (spun-up) such that the remaining model drift is significantly less than the signals under considerations or (3) both the latter and the former.

**Technical Issues**

**R: page 2, lines 13-14 & 27 : Work in progress for whom? The authors or the community in general, it could be either!**

A: We will change the phrasing.

**R: page 2, line 14 : Put comma after progress and delete comma from after extent.**

A: Thanks!

**R: page 2, line 18 : "As for now we know"?**

A: We will rephrase this expression in the revised version of the manuscript.

**R: page 3, line 11 "by chance", isn't it more by design?**

A: We did not expect that the high-resolution model configuration is so similar to the coarse resolution simulations we carried out so far. We expected substantial differences. These potential differences were the motivation to switch form coarse to high resolution. We will delete "by chance" in the revised version of the manuscript because it is confusing.

**R: page 3, line 13 : "allows to test" -> "allows us to test".**

A: Thanks!

**R: page 4, line 9, page 7, line 10 : Orphan lines like this pop up quite a bit throughout the paper.**

A: You are right! We will comb through and delete annoyances like these in the revised version of the manuscript.

**R: page 4, line 19 : 42.429.759 -> 42 429 759**

A: Yes.

**R: page 5, line 15 : I think the authors mean vicious cycle! (Even though viscous seems appropriate for numerical models)**

A: Ups, yes!

**R: page 5, line 24 : Galbreith -> Galbraith**

A: Sorry!

**R: page 8, line 5 : Spacial is an accepted alternative to the more common spatial spelling. However, the paper uses both spellings at different places.**

A: We will make it consistent.

**R: page 10, lines 21-24 : Is the reference to "module" meaning a specific part of the numerical code? or an inaccuracy in the overall representation of part of the system?**

A: We will clarify this in the revised version of the manuscript.

**R: page 12, line 16 : There's something wrong with the brackets here.**

A: We will correct this in the revised version of the manuscript.

**R: Figure 23 : noetig? in the caption.**

A: We will delete this in the revised version of the manuscript.

---

## Author Comment (AC2) · 12 Jul 2019

**General comments:**

**R: This paper presents a new ocean model configuration, which couples an ecosystem model to an active ocean-sea ice model, with a refined horizontal resolution in the Southern Ocean. The stated aim of this new configuration is to examine the Southern Ocean carbon budget and its sensitivity to wind changes. Although this new model configuration is promising, this paper fails to provide quantitative evidence of the added value of this configuration, and lacks adequate discussion of the applicability and limitations of the technical choices made or the resulting biases. Most results included are side-by-side plots of surface metrics in the simulation and observations. The model evaluation needs to include quantitative metrics and/or plots of differences, and would benefit from comparisons with existing reanalysis products or state estimates. Biases are only briefly (if at all) mentioned and there is generally no attempt to identify (and provide evidence for) the possible causes of these biases. In addition, the manuscript mentions a 'twin' configuration with coarser resolution, but does not include any comparison with that existing configuration (which would significantly improve the manuscript). The objectives of the wind sensitivity experiment are not clearly stated in the introduction, and the limitations of this experiment need to be discussed any time its results are included. This manuscript also needs to include a more detailed discussion about the applicability of the results and the limits of this configuration.**

**A:** To summarize, the reviewer proposes to add more quantitative estimates of model-data misfits and suggests to identify the possible causes of these biases.
In addition, he asks for a more comprehensive comparison between the coarse resolution simulation (which is not part of this GMD model documentation paper) and the high-resolution model simulation that is presented in this paper and suggests to include the objectives of the wind sensitivity experiment into the introduction.

We thank the reviewer for his constructive comments and we will add the respective information about the wind sensitivity experiment to the introduction (including its limitations).  Following his suggestions we will, further, add more quantitative estimates of model-data misfits to the revised version of the manuscript. Finally, we will put additional effort into identifying possible causes for model biases in the revised version of the manuscript.

As concerns a comprehensive comparison with the coarse resolution model: it is work in progress and will be presented in a forthcoming paper which references the high-resolution configuration presented in this GMD paper. We find that a comprehensive comparison is beyond the scope of this manuscript which merely should document the high-resolution model configuration and map out potential applications of it. We realize, however, that the manuscript - in its current form - raises wrong expectations. We will refocus accordingly in the revised version of the manuscript.

**Specific comments:**
**R: Abstract: The abstract makes the case that the model 1) combines a carbon inventory and eddy-permitting resolution in a way that outperforms existing models, 2) is suited to sensitivity experiments at decadal timescales, 3) can provide boundary condition to ice-sheet models. This manuscript would benefit from focusing the analysis and figures to demonstrate these 3 points, and from including a discussion section on the limitations of the model with regards to those 3 points.**

A: We agree with the reviewer that that the abstract is misleading in its present form and will revise it. The major point of the configuration is not that it outperforms existing models. The point is that, although eddy-resolving in the Southern Ocean and near-global in spatial extend, the remaining drift after the spinup in simulated dissolved inorganic carbon content is sufficiently small to study decadal carbon inventory variability. To our knowledge this is the first configuration/spinup of a "free" (i.e. not data-assimilated) model.

**R: Introduction: Overall, this section would benefit from a more focused approach, targeting specific dynamics or metrics that will be covered in the body of the paper.**

A: Agreed. Yes, this is confusing. We will make a major revision.

**R: Page 1, Line 20: There is no justification as to why 40S was chosen, or even simplyhow it relates to the above discussion about the location of the STF.**

A: Agreed. The only reason was to make comparisons with other papers using 40S easier. We will change this text. The other reviewer did not like it either ...

**R: Page 2, 29-31: Please consider how this fits within other work of Southern Ocean response to winds,such as (Spence et al, 2010).**

A: Thanks for the reference. We will use it in the respectively revised consideration.

**R: Page 3, 14: Where is this comparison between coarse and finer resolution model included? The manuscript would benefit from one.**

A: In our opinion a full coarse resolution / high resolution comparison in this manuscript will make the manuscript too extensive and is beyond our aim. Our aim here is to document the high-resolution model configuration settings, link them to open-accessable model output and provide a reference point for forthcoming studies (including those comparing different resolutions). The forthcoming studies will need

additional model evaluation because each scientific questions posed at a model typically asks for more "fit-for-the-purpose evaluation".

**R: Page 3,15-22: the discussion of the role of meltwater on ocean circulation or biogeochemistryis out of place here. Given that the current configuration does not include sub-ice-shelf cavities or parameterized meltwater fluxes, it will not be able to account for meltwater feedbacks. At best, this could be included in a discussion section to discuss options for further development.**

A: Agreed. We will put this more into the background.

**R: Page 4, line 15: Some justification on why topography needs to be filtered 3 times (even just a qualitative target) is needed here.**

A: Ideally, we would not have to filter at all. Numerical stability is guaranteed only for topographies so smooth that they barely represent actual conditions. A typical approach is to try with what little smoothing one can get away with. Turns out that 3 times is almost enough except for in a small confined area in the ACC where we had to apply additional viscosity. We will add a more formal explanation in the revised version of the manuscript.

**R:  Page 5, line 8-11:I have some concerns about the increase of viscosity over the Drake Passage. Some justification and possible root causes of this problem over Drake Passage need to be included.**

A: This relates back to your last issue with topographic filtering. We will provide a more comprehensive explanation on these issues in the revised version of the manuscript.

**R: Section 2.3 This section is too short and vague. If a change was made to the standard code and led to different results, please show the impact that this change has made (inan Appendix if necessary).**

A: We will add more information on this issue in the revised version of the manuscript. In a nutshell: sea ice attracted more sea ice until the model crashed. The state before the crash shows a LOT of sea ice in one spot and none elsewhere while the revised version shows a sensible distribution of sea ice. We tried dozens of different settings for the sea ice model - all to no avail. The problem persisted. We really got desperate and send our approach to the MOM-discussion forum. The response was that levitating sea ice is a viable option. We found also that this is used in ne NEMO community.

**R: Section 2.4 Design: please specify "complex-enough" for what? Or the model fidelity inrepresenting what? At least include the target variables and their scientific relevance.**

A: Agreed. We will revise the text accordingly.

**R: Why is the comparison with the coarse-resolution model not included in this paper?This manuscript needs to be self-sufficient.**

A: The aim of the manuscript is to document the high-resolution configuration. The coarse resolution configuration already published (https://www.biogeosciences.net/14/1561/2017/bg-14-1561-2017.html). We are sorry for the confusion. A thorough comparison has not been intended. We will state more clearly that we only intend to document our high resolution model configuration rather than raising the expectation that the paper contains a full high-res coarse-res model comparison.

**R: Section 2.6 Please include the frequency (or time scale) of boundary restoring. Also, the boundary restoring is necessary because the domain does not include dense water formation regions (it is not simply lack of processes).**

A: Agreed. We will add the respective information to the revised version of the manuscript.

**R: It is unclear what the wind experiment is supposed to address. If it is supposed to mirror the historical trend, please address why the physics are set to a historical trend while the carbon is maintained at a pre-industrial level. The objective of this experiment needs to be set up here, to provide context for the interpretation of results and discussion of their applicability.**

A: Agreed. We will add the respective information to the revised version of the manuscript

**R: Section 3. Why does this section not include any comparison with reanalysis data or state estimates? These datasets are commonly used in the literature, and providemore comprehensive data (sub-surface, time-varying...), which would help make a more compelling case for the fidelity of this new model configuration. As it is, thissection appears extremely light and unconvincing.**

A: The problem we have with reanalysis data and state estimates in this context is that they rely on very similar models than the one we present here. Here is an over-exaggeration of my problem: when comparing our model with a state estimate/reanalysis product in a region/time-interval where there are no observations, we essentially compare one model with another. Now, they could be similar without actually representing reality. So, in this specific case we find a comparison between our model and state estimate/reanalysis misleading.

**R: Section 3.1.1. This section lack qualitative evidence of the model's ability to represent the ocean circulation metrics mentioned.**

A: We will add more quantitative information in the revised version of the manuscript.

**R: It is necessary to define the meridionaloverturning as calculated here. Without a clear definition, the quantitative results are difficult to interpret or compare to other studies.**

A: Agreed, sorry, we will fix it (Reviewer 1 stumbled over the same issue).

**R: Section 3.1.2 This section lack qualitative evidence of the model's ability to produce 'realistic' levels of EKE. Given that this is a main point (included in the paper's title), putting plots side-by-side (Fig 8) is far from enough. Please include a plot showing the difference between the two datasets, and a quantification of mean EKE per region to provide some level of quantitative assessment.**

A: Thanks for the constructive comment. Will be done in the revised version of the manuscript.

**R: In addition, some discussion of the results is needed: what are some of the biases present? What may cause these biases? Finally, the conclusion that a 'realistic' level of EKE necessarily equates a good representationof eddy-driven processes is simplistic. Here, the only metric is surface EKE, where as eddy-driven processes, (including eddy-driven upwelling of nutrient-rich waters) occur over a range of depths. There is a vast body of literature investigating eddy processes in the Southern Ocean and what resolution may be necessary/sufficient to represent them adequately. Some discussion of this literature, and of how this particular model configuration fits in the context of other modeling studies is necessary.**

A: We will elaborate on this in the revised version of the manuscript.

**R: Section 3.1.3 Again, some quantitative comparison between the observation-baseddatasets and the modeled values is needed (e.g. plot of the difference), as well as a discussion of the biases (especially between the bottom temperature values, which has no discussion of biases at all). Combining Fig 10 and 11 would help the comparison.**

A: Thanks for the constructive comment. Will be done in the revised version of the manuscript.

**R: Section 3.1.5 A more comprehensive assessment of sea ice would make a more compelling case (e.g. sea ice concentration, sea ice thickness, annual cycle of sea ice area).**

A: We agree and will see what we can do (i.e. what observational products we can get).

**R: Section 3.1.6 This section needs a more comprehensive assessment of the model'sperformance in representing observed patterns of biogeochemical properties. A comparison to the Biogeochemical Southern Ocean State Estimate (B-SOSE) (Verdy andMazloff, 2017) would be a good step forward.**

A: I can understand the reviewer's push to promote B-SOSE because it is a good product. I do also agree that output from physical-biological data assimilation models are super valuable for a lot of purposes. In this specific case, however, I disagree: B-SOSE is based on a 1/3 degree (i.e. non-eddy resolving) model. Wherever there are no observation I would essentially compare my "free" eddy-resolving physical-biological data with a non eddy-resolving model. The breach in logic here is that: We do not know yet if non eddy-resolving models are sufficiently realistic. They may well be, but we do not know yet. The eddy-resolving configuration presented here will be used to work on this question. Please note that our only aim, for now, is to document the model settings of this configuration. We will make this more clear in the revised version of the manuscript and apologize for confusing phrasing in the original version of the manuscript.

**R: In addition, the biogeochemical performance of this model configuration should be shown (not just said) to be comparable to the one from the existing coarse resolution model. Likewise, there is little discussion of the possible causes of the biases, and lack of evidence to support the possible causes mentioned.**

A: O.K.

**R: Section 3.2 It is unclear what this sensitivity experiment is for, and why only the winds were changed.**

A: We will add the respective information in the revised version of the manuscript. (This section has been motivated by Lovenduski, N. S., Long, M. C., Gent, P. R., and Lindsay, K.: Multi-decadal trends in the advection and mixing of natural carbon in the Southern Ocean, Geophys. Res. Lett., 40, 139–142, doi:10.1029/2012GL054483, 2013.)

**R: It lacks discussion of the mechanisms leading to the change in over-turning circulation, or to the change of bottom water temperature. For the carbon results, it should be specified that the change in the experiment is showing only the**

**impact of the physical adjustment to winds (given that the carbon concentration is main-tained to pre-industrial levels).**

A: Agreed - we appear to promise more than we deliver. All we actually wanted to do is to sketch out potential applications of our configuration rather than presenting new science. We will adjust the text accordingly.

**R: Section 4: summary and conclusions This section makes qualitative statements about the model's fidelity, which have not been adequately supported by the body of the paper(similar to the abstract). As the model performance with respect to biogeochemistry is said to be similar to the coarser configuration, it is essential to demonstrate the benefits of this configuration with respect to eddy processes. Describing the configurationas a 'quantum leap' in modeling sounds over-reaching, given the lack of quantitative evidence included in the current manuscript to demonstrate the improved performanceof this model configuration compared to existing configurations. This section needs to include an in-depth discussion of the relevance and applicability of this model configuration. For example, the time scales examined here are the decadal timescales, while modeling of ice-shelf melt or Antarctic Bottom Water are more relevant to longer timescales.**

A: We apologize for the confusion. The "leap" refers to the fact that the "free" (i.e. not data assimilated in the Southern Ocean) eddy-resolving coupled ocean circulation biogeochemical model presented here has been successfully spun up such that the (spurious) trend in simulated Southern Ocean carbon content is small enough to study decadal variability and underlying mechanisms.

The way we presented the configuration seems to be misleading in that it implies that the fidelity of our model in terms of reproducing observations is a quantum leap. This is certainly not the case – the model performance is just what is to be expected from the current generation of these "free" models. Model configurations which apply data assimilation are - naturally - much closer to observations (even the coarser, non-eddy resolving).

The benefit of our "non-assimilated" configuration presented here is that it provides an additional tool to explore the interplay between atmospheric drivers, circulation and oceanic biogeochemistry. This interplay is of relevance because it affects processes of societal concern such as oceanic heat transport to ice-shelfs and oceanic carbon uptake. We thank the reviewer for highlighting this aspect and will clarify this issue.

**Technical Corrections:**

**A: We thank** the reviewer **for the time he put into identifying and listing the technical issues and his/her** constructive suggestions. We do not list them here - but, naturally, will take them all into account.

---

## Author Response (AR1)

Dear editor,

following the suggestions of the reviewers we herewith submit a revised version of the manuscript. Among our substantial changes is a new title: *MOMSO 1.0 - an eddying Southern Ocean model configuration with fairly equilibrated natural carbon*.

Major issues raised by the reviewers were lack of quantitative metrics of model-data misfits and a misguiding discussion of meridional overturning. In addition, the original manuscript apparently raised higher expectations than we could actually deliver in this model description paper.

Our changes to the manuscript include numerous additional quantitative measures, comparison with additional databases, a more comprehensive treatment of meridional overturning and a general overhaul dedicated to tone down the expectations raised with this model-description paper. In the course of this the number of figures and the length of the manuscript increased considerably (from 26 to 35 figures and 38 to 53 pages).

We hope that the revised version now meets the standard for a model description paper in GMD.

In any case, thank you for your work, effort and patience!

Kind regards,

Heiner Dietze (on behalf of the authors)

**Please note that this PDF-file continues with a point by point response to the reviewer's comments. The line numbers in these responses correspond to the revised manuscript version which tracks the changes and is attached at the end of this PDF-file.**

**Point by point response:**

**Please note that the line and page numbers in our responses refer to the revised version of the manuscript which tracks the changes to the text (in blue and red).**

------------------              **Anonymous Referee #1**              -----------------------------

**This paper documents the development of an interesting model configuration designed to allow for an eddying simulation of the Southern Ocean that also includes biogeochemistry. The model represents a series of compromises that mean whilst none of the components are ideal, the whole produces a potentially useful model simulation. The meat of the paper is validation of the physical circulation. This section requires some work, as there is considerable detail missing and a lack of quantification. The closing section of the paper looks at the biogeochemistry in the model's reference run and a short experiment with increased winds. This shows the capability of the model produce useful biogeochemical results, despite the physical circulation not being fully equilibrated. However, there is also a lack of detail here, which presents the full power of the modelling approach from being illustrated.**

Thank you for your constructive comments. We have added 8 new figures including Taylor diagrams (e.g. Fig. 22 and 28) which provide a means of quantifying model-data misfits. Further we extended the results section substantially (included the discussion of meridional sections and calculated a couple of metrics measuring the model-data misfit). The short experiment with increasing winds has been shifted into a new section titled *Research Questions* dedicated to roughly sketch out potential applications of the model.

**page 2, lines 4-9 : This is not an accurate representation of current thinking on theoverturning circulation of the Southern Ocean. There is no debate as to whether the bolus overturning due to eddy fluxes of thickness opposes the so-called Deacon cell. Itis extremely well established that it does so, to the degree that the Southern Ocean literature does not discuss the Deacon cell and instead talks about Eulerian overturningand bolus overturning. There is an on-going debate regarding the degree to which thebolus overturning cancels out the Eulerian, and that to which their residual respondst o forcing changes, particularly with respect to wind stress changes, and model differences. However, that this cancellations takes place is widely accepted. See, e.g.,Marshall & Radko (2003), Viebahn & Eden (2010), and Abernathey et al. (2011), etc**

We have added the respective citations (pg. 2, ln. 14). Further we explicitly state (pg. 12, ln. 4-6; pg. 19, ln. 12-13) that looking at Eularian overturning is not enough for a comprehensive analysis.

**page 2, line 24 onwards : There is at least one example of a fully-equilibrated, both thermodynamically and biogeochemically, model study that the authors could cite here;Munday et al. (2014). Whilst the model in Munday et al. is too coarse to be truly eddyresolving, it does have substantial internal variability and large-scale vortices, whichleads to changes in sensitivity to wind stress of the physical circulation consistent withhigher resolution models.**

We cite Munday at al. 2014 now on pg. 3, ln. 9.

**Section 2.1 : Also of importance is that, since z* is essentially an extension of the nonlinear free surface method of Campin et al. (2004) to all model levels, it also gives very accurate conservation of tracers.**

Added on pg. 5, ln. 5.

**page 5, lines 4-5 : The Smagorinsky scheme has a single coefficient, usually taken to be in the range 2-4, that allows some control over the viscosity. Please add the coefficient value you chose here. In addition, the issues arising near steep topography suggests that this coefficient was too small. Is there a reason why the authors chose to introduce additional viscosity instead of just increasing the viscosity value? Addition-ally, some of the issues could result from the use of purely Laplacian viscosity. Given the grid scale variance at the resolution of MOMSO, it is probably appropriate to start applying some biharmonic viscosity/diffusion to the model.**

Good point! Unfortunately, we could not muster up the resources to investigate this in depth. What we have done, though, is added the respective information (pg. 5, ln. 21-25) and concerns raised by the reviewer (pg. 6, ln. 9-10).

**page 5, lines 5-7 : The PPM scheme is a good choice for the advection of temper-ature/salinity. The choice of advection scheme for the biogeochemistry is also veryimportant (see Levy et al. 2001). Having different schemes for these two choices couldhave repercussions. Have the authors investigated this?**

We haven't. We added a bit of a discussion regarding this (pg. 5, ln. 26 - pg. 6, ln. 5).

**Section 2.5 : Initialising from a previous stratification/tracer distribution can be a very good way to shorten the spinup of high resolution models. Is there a reason why the temperature and salinity from the same run that produced the biogeochemical tracers wasn't also used?**

We wanted to initialize physics and biogeochemistry with observations but found that it is not recommended for one of the essential biogeochemical tracers. Hence, we decided to use coarse resolution model output for the biogeochemical tracers. We clarified our reasoning on pg. 7, ln. 9.

**Section 3.1 : A general criticism of this section is that it is largely qualitative, even when quantitative comparisons could be carried out, e.g. the spatial pattern and values of**

**EKE/SSH variance are available from the altimetry the authors use. A properly quantitative comparison would allow more detail to be drawn out. In addition, the authors highlight the surface velocities and EKE as having "remarkable" comparison to the altimetry. This is hardly remarkable; the surface velocities and EKE look largely as one would expect for a model of~1/5o grid spacing in the Southern Ocean (see, e.g. Del-worth et al., 2012, or Barnier et al., 2006, for examples of how similar resolution ocean models, or ocean components of coupled models, look). In addition, the deficiencies in particular comparisons are thoroughly glossed over. Something that stood out to me is that the flow is too zonal and there is too much EKE south of Australia. The East Australia Current is also poorly represented and the Agulhas retroflection has too small a region of high EKE. I'd expect the mean flow and eddies south of Australia to be related to the high temperatures south of this region shown in Figure 9. Most of the verification of the model is carried out on the surface quantities. Whilst this is appropriate given the length of the model run, which means that below the mixed layer there might not have been a great deal of adjustment, it is still useful to consider it briefly as it would place the later biogeochemical validation in context. At the very least, they should look at some temperature/salinity/density/biogeochemical transects across the Southern Ocean and the pycnocline depth. This will likely tie into the noted low value of the ACC transport. There is no discussion of the mixed layer depth. This is fairly well observed in recent years thanks to Argo floats and would be a key parameter for the exchange of carbon with the atmosphere and for the subduction of biogeochemical tracers from the surface.**

We calculated correlations and biases (e.g. pg. 11, ln. 12; pg. 12, ln. 18), and no longer refer to a "remarkable comparison" to altimetry. Instead we only make the point that we apparently do not underestimate mesoscale activity (pg. 12, ln. 24). In combination with a performance that is similar to coarse resolution models (c.f. Fig. 28) which are used to assess climate engineering options MOMSO provides an opportunity to compare the sensitivity of  coarse-resolution simulations with that of eddying model simulations.

Please note that the Australian Current and the Agulhas are already in the transition zone from high resolution to coarse resolution (i.e. they are north of 40S).

Following you advice we have added a discussion of surface mixed layer depth (pg. 12, ln. 29 - pg. 13, ln. 13, Fig. 22, Fig. 21) and meridional sections (Fig. 17- 20; e.g. pg. 13, ln. 23; pg. 14., ln. 25; pg. 16, ln. 14).

**Section 3.2 : Unfortunately, this section is completely deficient. There are many ways to calculate overturning (see discussion above about Eulerian vs. residual vs. bolus overturning) and there is no information given here as to what variety of overturning is presented in Figure 23. It should be some measure of the residual circulation, although given the increase in the WIND simulation it looks like it might be the Eulerian over-turning. A better comparison would be to calculate the residual overturning in density coordinates and compare the residual overturning, as well as the Eulerian and bolus overturning. This would give a much better sense of how the overturning is changing between the two experiments and how much the remaining drift in REF is affecting their results. If the subsurface stratification is changing, then it will make interpretation more difficult, since the there could be significant diapycnal transformations in the**

**streamfunction. This is another reason to look at the subsurface stratification earlier in the paper.**

We moved this section to pg. 18, ln. 22 section 4.1 and changed the context. The manuscript is a model-description paper and we only want to propose potential applications of the model rather than presenting a comprehensive analysis of overturning and eddy-compensation. We hope that this gets clearer now in Section 4 preluding what was formerly 3.2. Note also that we have added information/references regarding the different types of overturning (e.g. pg. 2, ln. 23 -14) and specify which kind of overturning we show (pg. 12, ln. 1).

**Minor Comments**
**page 1, line 20 : The choice of 40oS is not well justified, why not 35oS or 45oS? Thereare physical aspects of the circulation that could be cut off by this choice, for examplethe region in which the Agulhas current interacts with the ACC is very close to 40oS**.

Clarified on pg. 2, ln. 1-2.

**page 6, line 5-8 : Do the climatological estimates include the seasonal cycle?**

Clarified pg. 7, ln. 20.

**Section 3.1.4 : A missing quantitative comparison here is between the flux due to the surface restoring of salinity and the other surface fluxes that contribute to salinity changes.**

Agreed. Unfortunatly, we do not have archived the respective model output and could not muster up the resources to rerun the model. So this has to be discussed in forthcoming application/integrations of the model configuration and can not be part of this model-description paper.

**Figures such as 14 & 15, where the reader is invited to compare the REF and WIND experiments variation with the observed quantity would work much better as a single figure. Preferably as a single panel, with the observations as a third line.**

We were unsure how to deal with the different time periods.

**Section 3.1.6 : Based on Figures 19 & 20, the average concentration of both iron and phosphate appears low. Is this because the initial conditions for these fields are biasedlow? And if so, could the issues raised in this section be improved by simply adding more phosphate & iron?**

We have added some discussion of this starting on pg. 16, ln. 13. I suspect the answer is no. Too much iron and phosphate is leaving the Southern Ocean and not enough is coming back in.

**page 12, line 26 : In what sense to the compiler optimisations break reproducibility? Will the same model year, run on the same machine, be different if the model was rerun? Or will it only change when run on a different machine with different compilers? The first is rather alarming, the second would be very common.**

The second. It is a consequence of state-of-the-art compiler optimization which can take benefit of the "state" in which the computer is at the time of execution. Since the "state" (here referring to e.g. allocation of memory/cache by the operating system) differs from one second to another results also diverge. We have added a link to an intel document discussing reproducibility, floating point precision models and performance: pg. 18, ln 11.

**page 13, line 12 : Quantum leap is an overstatement, given the length of the physical model spinup, the still low resolution of the physical model and the documented deficiencies of both physical and biogeochemical system.**

Changed to "step forward" (pg. 20, ln. 16).

**Technical Issues**
Thank you! We changed them all following your suggestions.

**Please note that the line and page numbers in our responses refer to the revised version of the manuscript which tracks the changes to the text (in blue and red).**

------------------                Anonymous Referee #2          -----------------------------

**General comments: This paper presents a new ocean model configuration, which couples an ecosystem model to an active ocean-sea ice model, with a refined horizontal resolution in the Southern Ocean. The stated aim of this new configuration is to examine the Southern Ocean carbon budget and its sensitivity to wind changes. Although this new model configuration is promising, this paper fails to provide quantitative evidence of the added value of this configuration, and lacks adequate discussion of the applicability and limitations of the technical choices made or the resulting biases. Most results included are side-by-side plots of surface metrics in the simulation and observations. The model evaluation needs to include quantitative metrics and/or plots of differences, and would benefit from comparisons with existing reanalysis products or state estimates. Biases are only briefly (if at all) mentioned and there is generally no attempt to identify (and provide evidence for) the possible causes of these biases. In addition, the manuscript mentions a 'twin' configuration with coarser resolution, but does not include any comparison with that existing configuration (which would significantly improve the manuscript). The objectives of the wind sensitivity experiment are not clearly stated in the introduction, and the limitations of this experiment need to be discussed any time its results are included. This manuscript also needs to include a more detailed discussion about the applicability of the results and the limits of this configuration.**

The manuscript is intended to be a model description as opposed to a model-evaluation paper. That said: we added quantitative estimates, discussions and figures (among them, e.g. biases and correlations on pg. 11, ln. 11; pg. 12, ln. 18; taylor diagrams Fig. 22, 28)). The manuscript grew in length from 38 pages to 55. 8 figures have been added. We brought in more data products (sea ice dataset and surface mixed layer observations). Further we restructured the manuscript in order to make it clear that the intention of the twin configuration is simply to sketch out potential applications of the model rather than presenting a comprehensive study in the subject. Finally, we put our model performance into perspective by comparing it quantitatively with coarse resolution models (Sect. 3.04 - 3.07). Note that the intention of the model configuration is to provide an eddying circulation with equilibrated natural carbon. We deleted "quantum leap" and generally toned down in order not to raise expectations we could not not meet.

**Abstract: The abstract makes the case that the model 1) combines a carbon inventory and eddy-permitting resolution in a way that outperforms existing models, 2) is suited to sensitivity experiments at decadal timescales, 3) can provide boundary condition to ice-sheet models. This manuscript would benefit from focusing the analysis and figures to demonstrate these 3 points, and from including a discussion section on the limitations of the model with regards to those 3 points.**

We changed the abstract and hope now to raise lower expectations. These are: A description of a model with an eddying circulation with a relatively equilibrated natural carbon inventory. We

have restructured the results section and have now added Section 4: Research Questions which sketches two potential applications for which MOMSO - after further work - may serve as a starting point.

**Introduction: Overall, this section would benefit from a more focused approach, targeting specific dynamics or metrics that will be covered in the body of the paper**.

We refocused the introduction. The intent is now just to make clear that there are many open questions regarding the effects of eddies in the Southern Ocean and that this model configuration may be suited to add to the discussion because it has an eddying circulation. The intent of the manuscript is to describe the model configuration rather than to evaluate it.

**Page 1, Line 20: There is no justification as to why 40S was chosen, or even simplyhow it relates to the above discussion about the location of the STF.**

We added an additional explanation (pg. 2, ln. 1-2).

**Please consider how this fits within other work of Southern Ocean response to winds,such as (Spence et al, 2010).**

Added on pg. 3 ln. 14.

**Page 3, 14: Where is this comparison between coarse and finer resolution model included? The manuscript would benefit from one.**

We decided against a comprehensive coarse-res. versus high-res. comparison in this model description paper which already very long (53 pages). We also deleted the respective sentence from the introduction. Further we restructured the results section. The experiment with changing winds (which could be compared with the coarse resolution experiment in forthcoming studies) is moved to section 4 which is dedicated to suggest research questions which - after additional work - may be tackled with MOMSO.

**Page 3,15-22: the discussion of the role of meltwater on ocean circulation or biogeochemistry is out of place here. Given that the current configuration does not include sub-ice-shelf cavities or parameterized meltwater fluxes, it will not be able to account for meltwater feedbacks. At best, this could be included in a discussion section to discuss options for further development.**

Agreed. We deleted the respective paragraph from the introduction. Further, we explicitly state now that the configuration, in its present form, provides a starting point for future development (pg. 19., ln. 17).

**Page 4, line 15: Some justification on why topography needs to be filtered 3 times (even just a qualitative target) is needed here.**

We added an explanation on pg. 4, ln. 31 stating that we had good experience with such a procedure in the past. We agree that a more comprehensive analysis and the definition of quantitative criteria would be better. We would be grateful if the referee could steer us to relevant literature stating such criteria (so that we can put our procedure into perspective).

**Page 5, line 8-11: I have some concerns about the increase of viscosity over the Drake Passage. Some justification and possible root causes of this problem over Drake Passage need to be included.**

We agree that this is suboptimal. The justification is that this was the only way we found to continue the spinup. We added a description of a potential cause for it (pg. 6 ln. 9).

**Section 2.3 This section is too short and vague. If a change was made to the standard code and led to different results, please show the impact that this change has made (in an Appendix if necessary)**

We added some explanation and the exact change to the code on pg. 6, ln. 19.

**Section 2.4 Design: please specify "complex-enough" for what? Or the model fidelity in representing what? At least include the target variables and their scientific relevance. Why is the comparison with the coarse-resolution model not included in this paper? This manuscript needs to be self-sufficient.**

We elaborated on this on pg. 6, ln. 29. We also put the possibility to compare this high-resolution model configuration with a similar already-existing coarse resolution model configuration into the background of this model description paper.

**Section 2.6 Please include the frequency (or time scale) of boundary restoring. Also,the boundary restoring is necessary because the domain does not include dense water formation regions (it is not simply lack of processes). It is unclear what the wind experiment is supposed to address. If it is supposed to mirror the historical trend, please address why the physics are set to a historical trend while the carbon is maintained at a pre-industrial level. The objective of this experiment needs to be set up here, to provide context for the interpretation of results and discussion of their applicability.**

We added the fact that we restore to annual mean on pg. 7, ln. 20. The restoring timescale is listed on pg. 7, ln.27. We added a discussion on why we did experiment WIND on pg. 8 ln. 1-4.

**Section 3. Why does this section not include any comparison with reanalysis data or state estimates? These datasets are commonly used in the literature, and provide more comprehensive data (sub-surface, time-varying...), which would help make a more compelling case for the fidelity of this new model configuration. As it is, this section appears extremely light and unconvincing**

We substantially lengthened this section bringing in new data and analysis (discussion of meridional sections, seasonal cycle ice data, surface mixed layer data, discussion of biases and correlations, taylor diagrams). We now explicitly state that we want to have a performance similar to IPCC-type model (pg. 9 ln.1). We also state why we did not use state estimates (pg. 9, ln. 7-13).

**Section 3.1.1. This section lack qualitative evidence of the model's ability to represent the ocean circulation metrics mentioned. It is necessary to define the meridional**

**overturning as calculated here. Without a clear definition, the quantitative results are difficult to interpret or compare to other studies.**

We added quantitative metrics (pg. 11, ln. 10-14), specified the overturning presented (pg. 12, ln. 1) and highlight the relevance of other forms of overturning (pg. 12, ln. 4-6).

**Section 3.1.2 This section lack qualitative evidence of the model's ability to produce 're-alistic' levels of EKE. Given that this is a main point (included in the paper's title), puttingplots side-by-side (Fig 8) is far from enough. Please include a plot showing the difference between the two datasets, and a quantification of mean EKE per region to providesome level of quantitative assessment. In addition, some discussion of the results isneeded: what are some of the biases present? What may cause these biases? Finally, the conclusion that a 'realistic' level of EKE necessarily equates a good representation of eddy-driven processes is simplistic. Here, the only metric is surface EKE, whereaseddy-driven processes, (including eddy-driven upwelling of nutrient-rich waters) occurover a range of depths. There is a vast body of literature investigating eddy processes in the Southern Ocean and what resolution may be necessary/sufficient to represent them adequately. Some discussion of this literature, and of how this particular model configuration fits in the context of other modeling studies is necessary**

We toned-down our claims. The title now reads "an eddying Southern Ocean model configuration" and pg. 12, ln. 24 now merely claims that we apparently do not underestimate EKE in the high-resolution model domain (rather than advertising remarkable fit to observations). Further we added a discussion of biases and correlations (pg. 12, ln. 15-23) and the additional Fig. 9 shows now the difference between model and observations.

**Section 3.1.3 Again, some quantitative comparison between the observation-based datasets and the modeled values is needed (e.g. plot of the difference), as well as a discussion of the biases (especially between the bottom temperature values, which has no discussion of biases at all). Combining Fig 10 and 11 would help the comparison.**

Now starting on pg. 13, ln. 15 the respective section has been extended. It features now a discussion of water masses which includes temperature biases. We were unsure how to combine Fig. 10 and 11 because of the differing time axis. Regarding bottom water temperatures: we have shifted this application of the mode into the background. It is now moved to section 4.2 on pg 19. which is preluded by the intro to section 4 on pg. 18 reading: "The purpose of this section is to outline research questions for which MOMSO *may* serve as a tool."

**Section 3.1.5 A more comprehensive assessment of sea ice would make a more com-pelling case (e.g. sea ice concentration, sea ice thickness, annual cycle of sea icearea).**

Following Russell 2008 we brought in a new dataset and added Figure 26 showing a comparison between modeled and observed annual cycle of sea ice area.

**Section 3.1.6 This section needs a more comprehensive assessment of the model's performance in representing observed patterns of biogeochemical properties. A comparison to the Biogeochemical Southern Ocean State Estimate (B-SOSE) (Verdy and Mazloff, 2017) would be a good step forward. In addition, the biogeochemical performance of this model configuration should be shown (not just said) to be comparable to the one from the existing coarse resolution model. Likewise, there is little discussion ofthe possible causes of the biases, and lack of evidence to support the possible causesmentioned.**

We toned down the expectations. Our goal is now to be of similar fidelity than coarse resolution IPCC-type models (because we plan to compare the sensitivity of IPCC type models to our high-resolution model which we merely intend to describe her in this model-description paper). We added a discussion of meridional sections of nutrients and oxygen (starting on pg. 16, ln. 14) and included a Taylor diagram (Fig. 28) which shows a comparison with the coarse resolution model. Please note that the coarse resolution model is no longer such an integral part of this paper which describes a high-resolution model configuration (the respective references to the coarse resolution model have been shifted into a new section titles research questions on pg. 18, ln. 14 dedicated to roughly sketch out questions for which MOMSO may serve as a tool - after additional work).

**Section 3.2 It is unclear what this sensitivity experiment is for, and why only the windswere changed. It lacks discussion of the mechanisms leading to the change in over-turning circulation, or to the change of bottom water temperature. For the carbon results, it should be specified that the change in the experiment is showing only the impact of the physical adjustment to winds (given that the carbon concentration is maintained to pre-industrial levels).**

Sorry. We agree that the structure of the paper was confusing. We moved this section now to Section 4: research questions where we map out research questions for which MOMSO may turn out to be useful. We added an explanation of why these questions would be of interest.

**Section 4: summary and conclusions This section makes qualitative statements about the model's fidelity, which have not been adequately supported by the body of the paper(similar to the abstract). As the model performance with respect to biogeochemistry is said to be similar to the coarser configuration, it is essential to demonstrate the benefits of this configuration with respect to eddy processes. Describing the configuration as a 'quantum leap' in modeling sounds over-reaching, given the lack of quantitative evidence included in the current manuscript to demonstrate the improved performance of this model configuration compared to existing configurations. This section needs to include an in-depth discussion of the relevance and applicability of this model configu-ration. For example, the time scales examined here are the decadal timescales, while modeling of ice-shelf melt or Antarctic Bottom Water are more relevant to longer timescales.**

We toned down from "quantum leap" to "step forward". We merely state now (already in the title) that the configuration is eddying and has a natural carbon inventory that is fairly

equilibrated so that decadal-scale changes to the wind are clearly recognizable from the still-persistent background drift.

**Technical Corrections:**

Thank you! We changed them all following your suggestions.

[revised manuscript text omitted]